# A comparison between humans and AI at recognizing objects in unusual poses

**Netta Ollikka**                                                     *netta.ollikka@aalto.fi*
*Department of Neuroscience and Biomedical Engineering*
*Aalto University, Espoo, Finland*

**Amro Abbas**                                                        *afagiri@aimsammi.org*
*The African Institute for Mathematical Sciences, Mbour-Thies, Senegal*

**Andrea Perin**                                                      *xinandre@gmail.com*
*Department of Computer Science*
*Aalto University, Espoo, Finland*

**Markku Kilpeläinen**[†]                                             *markku.kilpelainen@helsinki.fi*
*Department of Psychology and Logopedics*
*University of Helsinki, Finland*

**Stéphane Deny**[†]                                                  *stephane.deny.pro@gmail.com*
*Department of Neuroscience and Biomedical Engineering*
*Department of Computer Science*
*Aalto University, Espoo, Finland*

**Reviewed on OpenReview:** *https://openreview.net/forum?id=XXXX*

## Abstract

Deep learning is closing the gap with human vision on several object recognition benchmarks. Here we investigate this gap in the context of challenging images where objects are seen in unusual poses. We find that humans excel at recognizing objects in such poses. In contrast, state-of-the-art deep networks for vision (EfficientNet, SWAG, ViT, SWIN, BEiT, ConvNext) and state-of-the-art large vision-language models (Claude 3.5, Gemini 1.5, GPT-4, SigLIP) are systematically brittle on unusual poses, with the exception of Gemini showing excellent robustness to that condition. As we limit image exposure time, human performance degrades to the level of deep networks, suggesting that additional mental processes (requiring additional time) are necessary to identify objects in unusual poses. An analysis of error patterns of humans vs. networks reveals that even time-limited humans are dissimilar to feed-forward deep networks. In conclusion, our comparison reveals that humans are overall more robust than deep networks and that they rely on different mechanisms for recognizing objects in unusual poses. Understanding the nature of the mental processes taking place during extra viewing time may be key to reproduce the robustness of human vision *in silico*. All code and data is available at `https://github.com/BRAIN-Aalto/unusual_poses`.

## 1 Introduction

In an era marked by the rapid advancement of deep learning for computer vision, a natural question arises: Can machines meet or even exceed the capabilities of the human visual system? Numerous recent studies have shown that deep networks outperform humans on well-known object recognition benchmarks (e.g., ImageNet: He et al. (2015); Vasudevan et al. (2022); Dehghani et al. (2023)). Deep networks are even said to outperform

---

[†]equal contribution as last authors.

humans on *out-of-distribution* recognition tasks (Geirhos et al., 2021), where the images used for evaluation are subjected to distortions that the networks were not specifically trained on.

However, studies that investigate out-of-distribution generalization mostly focus on local distortions (e.g., texture modifications, blur, color modifications) as opposed to transformations that affect the global structure of the image, such as a change in viewpoint. A few studies have investigated the generalization capability of deep networks to recognize objects in various poses, both in non-adversarial (Alcorn et al., 2019; Ibrahim et al., 2022; Madan et al., 2022; Abbas & Deny, 2023) and adversarial settings (Zeng et al., 2019; Madan et al., 2021b;a), and they show a substantial degradation of network performance for unusual poses. However, a direct comparison with human vision has been missing from these studies, leaving open the question of whether humans outperform networks on this task.

Here, we compare human subjects with state-of-the-art deep networks for vision and state-of-the-art large vision-language models at recognizing objects in various poses. We show that—while both humans and networks excel on upright poses—humans substantially outperform networks on unusual poses (except for Google's Gemini showing exceptional robustness in that condition). As we limit viewing time, humans exhibit a similar brittleness to deep networks on unusual poses, demonstrating the necessity of additional mental processes for analysing these challenging images. Although networks and time-limited humans are both brittle, an analysis of their patterns of error reveals that time-limited humans make different mistakes than networks. In conclusion, our results show that (1) humans are still much more robust than most networks at recognizing objects in unusual poses, (2) time is of the essence for such ability to emerge, and (3) even time-limited humans are dissimilar to deep neural networks.

## 2 Methods

### 2.1 Dataset collection

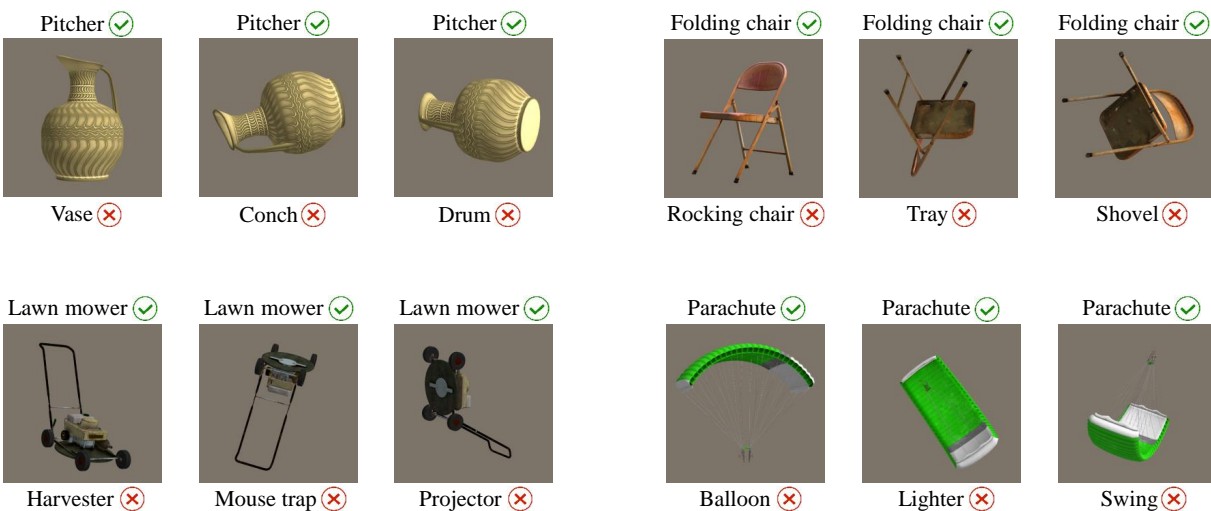

Figure 1: **Example images of the dataset, and corresponding answer choices.** Four examples of objects and their three different rotations: *(left)* upright, *(middle)* rotated and correctly labeled by EfficientNet, and *(right)* rotated and incorrectly labeled by EfficientNet. Above each image is shown the correct label and below each image is the alternative label that we selected based on EfficientNet's predictions (see *Dataset collection* 2.1 for details of this selection).

We collected a dataset of objects viewed in different poses (upright and rotated out-of-plane), to test the ability of humans to recognize these objects, and compare this ability to state-of-the-art deep networks (Figure 1).

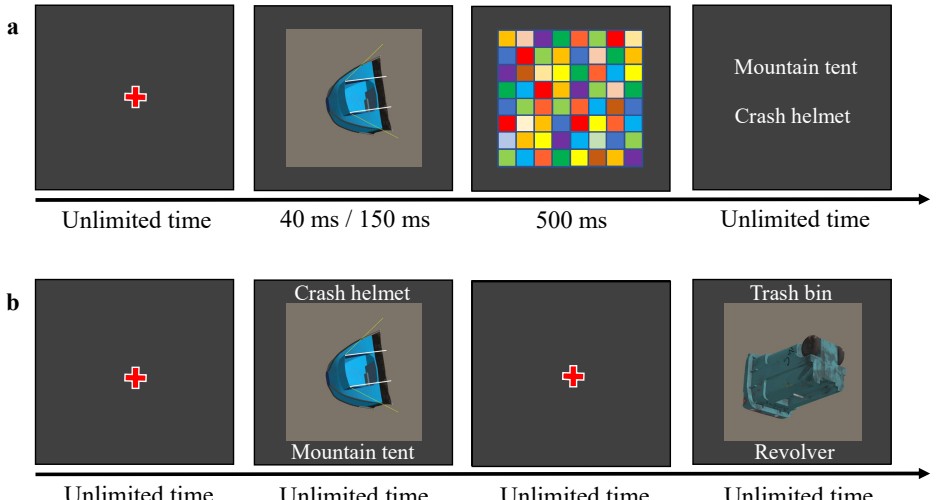

Figure 2: **Description of the human tests used in this study. a)** Task with limited viewing time: First the subject fixates on a cross, then an image is displayed for either 40 ms or 150 ms, followed by a dynamic checkerboard mask shown for 500 ms. Then the subject is asked to choose between two labels for the image (i.e., two-forced-choice task), and has an unlimited time to answer. **b)** Task with unlimited viewing time: similar test-setting, but now the image and the answer choices are displayed together for an unlimited viewing time and without back-masking.

We chose 51 different object categories from the ImageNet classes (see *Appendix* D for the list of objects), and obtained a corresponding 3D model for each category from Sketchfab (https://sketchfab.com/). The objects were chosen based on a few criteria:

- Objects should look distinct in different rotation angles, e.g., a symmetrical ball wouldn't be an acceptable object.

- Objects should have clear upright poses, e.g., a pen wouldn't be an acceptable object.

- The object in its upright pose should be correctly labeled by Noisy Student EfficientNet-L2, pretrained on JFT-300M and finetuned on ImageNet (Xie et al., 2020) (referred to as EfficientNet below). We chose EfficientNet to guide our selection of objects because it is the best open-source network at recognizing objects in unusual poses, according to a recent comprehensive study comparing 37 state-of-the-art networks on this task (Abbas & Deny, 2023).

Once the objects that satisfy the criteria were chosen, we rendered 180 different random rotations of each one of them (along axis x, y and z). These rotations were given to EfficientNet to be labeled. Then, we performed a selection of poses. First, we selected unusual poses that EfficientNet was able to correctly identify. Second, we selected unusual poses that EfficientNet failed to correctly identify. The dataset thus consisted of three different types of poses: (1) upright, (2) rotated and correctly identified by EfficientNet (rotated-correct condition), and (3) rotated and incorrectly identified by EfficientNet (rotated-incorrect condition).

Once the different poses for each object were chosen, we created two-forced choice questions for each image based on EfficientNet's predictions. The choices consisted of the correct label, and an incorrect label. In the cases where EfficientNet was able to correctly identify the object, we chose the second best guess of EfficientNet to be the incorrect label, as measured by the category corresponding to second most activated unit of the output layer of EfficientNet.[1] In the cases where EfficientNet incorrectly predicted the label, we

---

[1]In some cases, we had to choose the 3rd or the 4th best guess of EfficientNet, the criterion to discard the 2nd best guess being that it would be too similar to the correct label for a non-specialist human (e.g., golden retriever vs. labrador).

used the wrong predicted label as the incorrect label. The final dataset contained 147 different images: 51 upright poses, 51 rotated-correct poses, and 45 rotated-incorrect poses (examples in Figure 1).[2]

**Caveat 1:** We used EfficientNet to identify challenging object poses and label alternatives. A caveat of this selection method is that it may introduce a bias against networks, since half of the images were selected to fool EfficientNet (which is a neural network). To mitigate this bias, we separately compare networks and humans on the half of images that have *not* failed EfficientNet (rotated-correct condition in Fig. 4). Essentially, our conclusions remain identical on that subset: humans (nearly 100% accurate) are much more robust than pure-vision deep networks (overall best network of our collection: SWAG: 82% accuracy) and of most vision-language models (except Gemini). Additionally, we note that our selection is hardly adversarial, since EfficientNet failed on a large fraction of random poses (14.5% of poses, cf Abbas & Deny (2023)). Our study is thus not akin to studies performing fine pose optimization to find adversarial examples (e.g., Madan et al. (2021a)).

**Caveat 2:** The stimulus set applied here is not entirely ecologically realistic, as all objects are synthetic 3D models and are displayed without a background. However, these choices are justified by our objective to create a highly controlled stimulus set and experimental setup: performance in canonical view is compared with performance in a rotated view of the very same object, both shown without background. Thus, any performance differences between these conditions, be it human or network, can only be caused by rotation. Conversely, any robustness found across conditions can only be due to rotation-invariant recognition, not context-based reasoning.

## 2.2 Psychophysics experiments

### 2.2.1 Observers

Altogether 24 observers (12 women, 12 men, aged 19-54) participated in the experiments. Our observers comprised Finnish students from diverse academic backgrounds, including disciplines such as engineering, psychology, business, and medicine. Participants were required to have normal vision (i.e., individuals using vision aids such as glasses to achieve normal vision) or corrected-to-normal vision. Additionally, the observers did not report any medical condition (epilepsy, migraine, etc.) that could affect the results.

### 2.2.2 Apparatus and stimuli

The object images were synthetic images of objects belonging to the ImageNet database classes. For information on the process of stimulus creation, see *Dataset collection* 2.1 above. Noise masks, which were presented after object image presentation, consisted of dynamic white noise chromatic checkerboards (0.67 degrees of visual angle check size, 100 Hz temporal frequency). The diameter of the object images and the noise masks were 13.3 degrees of visual angle. The stimuli were presented using MATLAB Psychophysics Toolbox in a 22.5" VIEWPixx display with a resolution of 1200 x 1920 pixels, a frame rate of 100 Hz, and a viewing distance of 54 cm.

### 2.2.3 Procedure

Prior to the experiments, observers were provided with a small training set, in order to familiarize themselves with the experimental procedure. None of the training objects were featured in the real experiment. After this training, each observer participated in a *limited viewing time* experiment and an *unlimited viewing time* experiment. In the *limited viewing time* experiment, each trial proceeded as follows (see Figure 2a). First, a fixation cross-hair was presented in the centre of the screen. Once the observer was ready, they pressed the space bar. After 500 ms, the object image was presented. The duration of the object image presentation was 40 ms for 12 observers, and 150 ms for the other 12. These stimulus durations (40 and 150 ms) were based on pilot experiments which suggested that those durations would lead to performance levels between perfect and chance level performance. The object image was immediately followed by 500 ms of dynamic chromatic

---

[2]There were fewer incorrectly identified poses, because in six cases, EfficientNet was able to correctly identify all 180 rotated poses.

white noise. After that, two response alternatives (in Finnish), one of them always the correct one, were presented and the observer had to choose (by pressing one of two keys on the keyboard) which alternative corresponded to the presented object. Each observer performed 49 trials, in which the image was in one of three types of poses: upright in 17 trials, rotated-correct (correctly classified by EfficientNet, see *Dataset collection* 2.1) in 17 trials, and rotated-incorrect (incorrectly classified by EfficientNet) in 15 trials. The trial types were interleaved and presented in random order. The *unlimited viewing time* experiment (see Figure 2b) followed the same structure, except that the image remained on the screen until the observer responded (2.0 s on average), the labels were present from the start, and no noise was presented after the object image.

Each observer saw each object only in one pose during the limited viewing time experiment. The object-pose combinations were, however, balanced across observers such that each object was shown in a particular pose (e.g., upright) an equal amount of times. The order in which the objects were presented was different for every subject in order to avoid potential order effects. Furthermore, each observer saw exactly the same object-pose combinations they had seen in the limited time experiment for the unlimited time experiment. This allowed more powerful within-subject statistical analyses. In principle, the previous experience with the same images could affect the observers performance in the unlimited viewing time experiment. However, since the second presentation was not time-limited, making the task very easy for the viewer (see Results), we expect any advantage of having the image flashed previously to be negligible.

## 2.3 Machine tests

We then implemented the same tests on a collection of 5 state-of-the-art deep networks for vision (referred below as 'pure vision networks') and 6 large vision-language models (VLM). Since EfficientNet was used to select the problematic images and labels, we did not include EfficientNet in this collection nor in our study of network robustness.

We tested the 5 following pure vision networks: SWAG-RegNetY (Singh et al., 2022), BEiT-L/16 (Bao et al., 2021), ConvNext-XL (Liu et al., 2022), SWIN-L (Liu et al., 2021), and ViT-L/16 (Dosovitskiy et al., 2020) (see *Appendix* C for model descriptions). The network selection was based on previous work from Abbas & Deny (2023), which thoroughly evaluated the robustness of 37 state-of-the-art networks to objects presented in unusual poses. This collection provides a diverse set of state-of-the-art and very large models trained on large datasets (e.g., Imagenet-21K, Instagram 3.6B) with various architectures (ConvNet and Transformers) and training objectives (supervised and self-supervised). These networks obtained some of the most robust results to unusual poses according to Abbas & Deny (2023), and all of them were fine-tuned to the categories of ImageNet.

We mirrored the experiment done on humans by submitting each pure vision network to the same two-forced-choice test. Each network was shown the 147 different images of objects, 51 upright poses, 51 rotated-correct poses, and 45 rotated-incorrect poses. We compared the activations of the units of the final layer corresponding to each of the two categories given as choices for each image, and selected the category with corresponding highest unit-activation among the two to be the chosen answer.

Our selection of VLMs included Gemini 1.5 Flash, Gemini 1.5 Pro, Claude 3 Opus, Claude 3.5 Sonnet, GPT-4o, GPT-4-vision-preview and SigLIP (Zhai et al., 2023). For the large-language models (all models excluding SigLIP), the experiment was conducted via the API. Each model was shown the 147 images and provided with the following prompt (see examples in Appendix A):

*What's in this image?*
*A. [label 1]*
*B. [label 2]*
*Choose either A or B and answer in one or two words.*

If the model chose the correct label, the answer was counted as correct. If the model chose the wrong label or gave a response outside of the label options, the answer was counted as incorrect. If the model didn't provide any answer (e.g., Gemini threw a safety error for a few images), the image was disregarded and not included in the analysis.

For SigLIP, a state-of-the-art contrastive text-image model (successor of CLIP (Radford et al., 2021)), we fed it the image as well as the two label options. The label with the highest score of similarity was accepted as the answer of the model.

## 2.4 Statistical tests

**T-tests:** We conducted a series of t-tests to compare the accuracies of different observers (networks and/or humans) and of the same observers in different viewing conditions (e.g., upright vs. rotated objects, limited vs. unlimited viewing time). When the data came from the same observers in two different viewing conditions, we used paired t-tests (i.e., 40 ms upright vs. 40 ms rotated) and unpaired t-tests when the observers were different (i.e., 40 ms rotated vs. 150 ms rotated). T-tests were run on percentages of correct answers.[3]

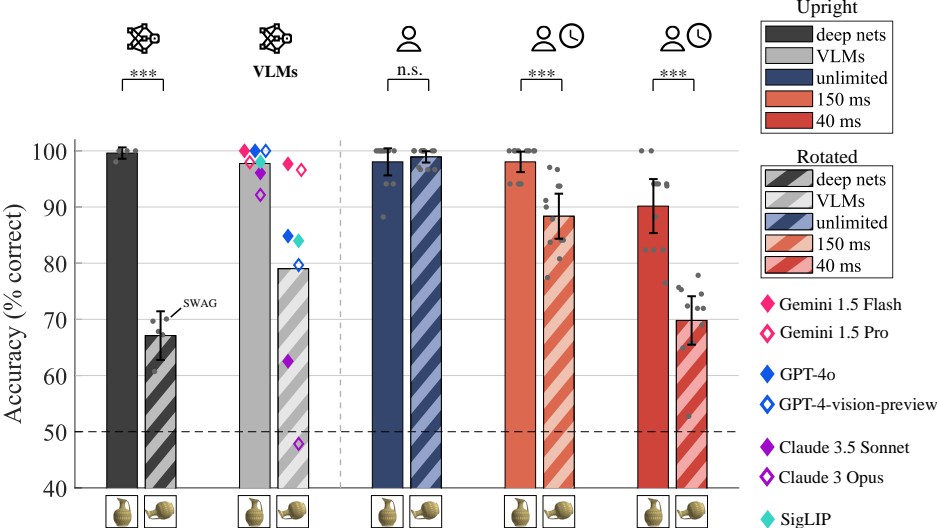

Figure 3: **Comparing neural networks and humans at recognizing objects in various poses.** *Dark grey bars*: Average performance of pure vision deep networks (left: upright vs. right: rotated) with 95% confidence intervals (n=5, grey points represent individual network performances). *Light grey bars*: Average performance of seven large vision-language models (VLMs). Diamonds indicate individual model performances (full diamonds show the best-performing version of the different model classes). *Blue bars*: Average human performance with unlimited viewing time (n=12, grey points represent individual performances). *Orange & Red bars*: Average human performance with limited viewing time (150 ms and 40 ms, respectively, n=12, grey points represent individual performances). Chance performance is 50%. Three stars (***) indicate highly significant differences (p<0.001), "n.s." for not significant. With unlimited time, humans excel at recognizing rotated objects, while pure vision networks struggle (best: SWAG at 70.1%). GPT-4, Claude and SigLIP models follow the same pattern as the pure vision networks, with a significant drop in accuracy for rotated images compared to upright ones. However, Gemini 1.5 Flash mirrors human performance with unlimited viewing time, achieving 97.7% accuracy on rotated images compared to human accuracy of 98.9%. Limiting human viewing time (40 ms or 150 ms) impairs their ability to recognize rotated objects, substantially more than upright objects, bringing their performance closer to network levels.

**Error consistency:** When two observers $i$ and $j$ (networks and/or humans) respond to the same $n$ trials, we can measure how well their responses (correct/incorrect) align by computing their observed error overlap

---

[3]We acknowledge that %-accuracy is a non-linear index of performance and according to signal detection theory (Green et al., 1966), z-scores would be a more appropriate measure to use when computing means and performing statistical tests. However, since %-accuracy is more commonly used in the field of machine learning, we used %-accuracy when performing t-tests. We have, however, also conducted all our statistical tests with z-scores (not reported) and the changes in results were very minor, with no effect on the conclusions of this study.

$c_{obs_{i,j}}$:

$$c_{obs_{i,j}} = \frac{e_{i,j}}{n}, \tag{1}$$

where $e_{i,j}$ is the number of same responses, either both correct or incorrect. However, considering only the observed error overlap has limitations, because this overlap is expected to depend on overall accuracy of the observers in virtue of the probability of coincidence of two independent binomial variables (e.g., a network and a human with 99% performance are expected to have more overlap than a network and human with 90% performance). To address this issue, we compare observers $i$ and $j$ to a theoretical model of independent binomial observers. This model considers two observers making random decisions, and thus we can only expect overlap due to chance. This expected error overlap $c_{exp_{i,j}}$ is given by:

$$c_{exp_{i,j}} = p_i p_j + (1 - p_i)(1 - p_j), \tag{2}$$

where $p_i p_j$ is the probability that observes give the same *correct* response by chance and $(1 - p_i)(1 - p_j)$ the probability that they give the same *incorrect* response by chance. To evaluate the consistency between the two observers $i$ and $j$, and determine whether it goes beyond what could be expected by chance, we use a metric called error consistency (Geirhos et al., 2020; 2021) (i.e., Cohen's $\kappa$) given by:

$$\kappa_{i,j} = \frac{c_{obs_{i,j}} - c_{exp_{i,j}}}{1 - c_{exp_{i,j}}}. \tag{3}$$

Error consistency compares observed consistency to expected consistency, and allows us to quantify whether the observed consistency is larger than would have been expected by chance.

## 3 Results

We compared human and machine vision on a task consisting in recognizing objects in unusual poses. For this, we selected 5 different pure vision deep neural networks (SWAG, ViT, SWIN, BEiT, ConvNext), chosen for their state-of-the-art performance in recognizing objects in unusual poses (Abbas & Deny, 2023), and 7 state-of-the-art large vision-language models (Gemini 1.5 Flash, Gemini 1.5 Pro, Claude 3 Opus, Claude 3.5 Sonnet, GPT-4o, GPT-4-vision-preview, SigLIP). We performed a two-alternative-forced-choice object categorization task: an image of an object was presented to the viewer (either deep network or human participant), and then the viewer had to choose between two different names (i.e., labels) for the object. The two labels were selected based on the highest activations of Noisy Student EfficientNet's output layer (Xie et al., 2020), the most robust network to unusual poses according to Abbas & Deny (2023). For pure vision networks, the choice was made by looking at the highest activation of the softmax output layer for these two labels. EfficientNet was excluded from our analyses comparing humans and networks because images and labels were chosen based on its mistakes (see Methods). For the vision-language models, each VLM was given an image and prompted with the 2 label choices, and the label chosen by the model was selected to be its answer. The performance of VLMs is discussed in the last paragraph of results, while previous paragraphs concentrate on the comparison of pure vision networks and humans.

**With unlimited viewing time, humans outperform deep networks at recognizing objects in unusual poses.** First, we compared the ability of humans and networks to recognize objects in upright and unusual poses when humans were not given any time limit for the object recognition (Figure 3). In the upright poses, both humans and neural networks performed very well: 4 networks obtained 100% accuracy and one (ViT-L) failed to correctly label only 1 out of 51 objects; humans had slightly lower performance with an accuracy of 98.0% ± 2.4% (95% confidence intervals). The very few errors that human observers made are likely to be keystroke errors and misunderstandings of the verbal response alternatives (as reported by some subjects). We conducted an additional test to discern whether network errors arise from changes in viewpoint, or alterations in image statistics between training and testing distributions (natural vs. synthetic). We subjected all five of our networks to a dataset comprising objects rotated at angles of ±10° from their canonical poses. All of our networks obtained perfect or near-perfect results (SWAG-RegNetY: 100%, BEiT-L/16: 99.3%, ConvNext-XL; 98.2%, SWIN-L: 91.1%, ViT-L/16: 95.0%). *For unusual poses, human participants with unlimited viewing time clearly outperformed neural networks.* For humans, the performance on rotated

images and upright images was on par (rotated condition: 98.9% ± 1.0%)(t(11) = -0.79, p = 0.45), whereas for networks, the performance drastically dropped between upright (99.6% ± 1.0%) and rotated objects (67.1% ± 4.3%), with a drop of over 30% performance on average (t(4) = 18.7, p = 4.8e-05). Even the most robust network of our collection, SWAG trained on IG-3.6B (Singh et al., 2022), showed a drop in performance of 30% (from 100% on upright poses to 70.1% on rotated poses).

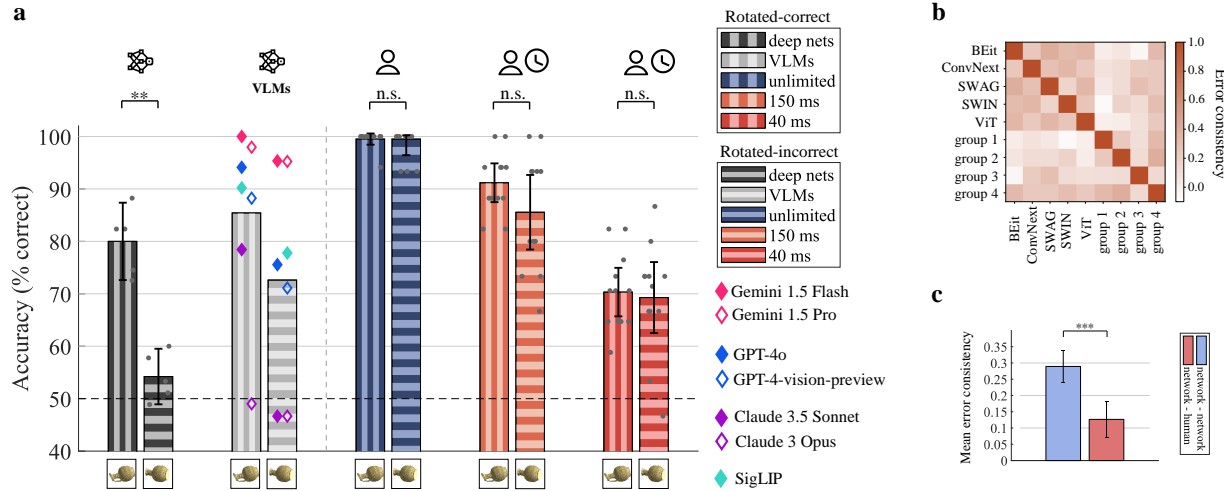

Figure 4: **Error patterns are different for neural networks and time-limited humans. a)** Comparison of human and network accuracy for the rotated-correct condition (EfficientNet was correct on these rotations) vs. rotated-incorrect condition (rotations that have failed EfficientNet). Humans show consistent accuracy between rotated-correct and rotated-incorrect conditions. In contrast, networks, including the seven VLMs (represented by diamonds), exhibit a performance drop. However, for Gemini 1.5 Flash, Gemini 1.5 Pro and Claude Opus, the decrease in performance between the two conditions is not as notable. **b)** Error consistency analysis performed on the 5 neural networks (not including VLMs) and 40 ms time-limited human subjects. 12 human subjects were partitioned into 4 groups of 3, so that every group saw each rotated image exactly once. The darker red cluster for networks (dark red = highly consistent errors) indicates that they have similar patterns of error, which are not shared by human subjects, highlighting that EfficientNet errors transfer better to other networks than to humans. **c)** Mean error consistencies were calculated by comparing networks with each other and with human subjects (i.e., average computed over the different matrix clusters). A two-tailed unpaired t-test confirms that networks indeed make more consistent errors with each other than with humans (t(28) = 4.0, p = 3.8e-04).

**Under limited viewing time, humans are faulty on unusual poses, like deep networks.** In the unlimited viewing time condition, observers used on average 2 seconds before taking a decision. We next studied the performance of human observers when the viewing time was limited to 40 ms and 150 ms followed by a dynamic white noise chromatic mask (Figure 3) (see *Procedure* 2.2.3 for the justification for these duration choices). With the upright poses, the accuracy of human observers decreased quite modestly (to 90.2% ± 4.8%) when viewing time was 40 ms. The unlimited vs. 40 ms difference is statistically significant (t(22) = 3.19, p = 0.0042). The performance for 40 ms rotated images (69.8% ± 4.3%) drastically decreased compared to 40 ms upright images, with a drop of 20% (t(11) = 8.5, p = 3.7e-06). Additionally, the performance on 40 ms rotated images notably decreased compared to unlimited viewing time (98.9% ± 1.0%)(t(22) = 14.3, p = 1.2e-12). At 150 ms exposure time, performance was barely affected when the objects were presented upright (98.0% ± 1.8% for the 150 ms viewing condition compared to 98.0% ± 2.4% for the unlimited viewing time condition). Subjects performed substantially better at recognizing objects in unusual poses when given 150 ms (88.4% ± 4.0%) compared to 40 ms (69.8% ± 4.3%)(t(22) = 6.9, p = 6.7e-07), but not as well as with unlimited viewing time (98.9% ± 1.0%)(t(11) = 7.0, p = 2.3e-05). We conclude that the process needed to

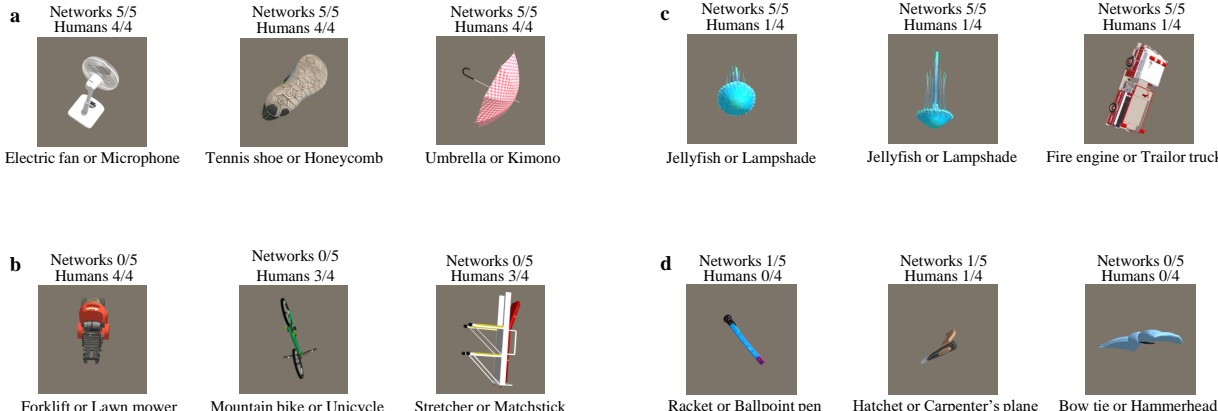

Figure 5: **Examples of objects in unusual poses, where deep networks for pure vision and 40 ms time-limited humans made similar and differing errors.** Above each image is a quantitative score of correct answers and below each image are the given answer choices, correct answer being the first. **a)** Images where all networks and humans correctly labeled the object. **b)** Images where all networks failed to correctly label the objects, but where humans mostly chose the correct answer. **c)** Images, where all the networks correctly labeled the objects, but where most humans failed. **d)** Images, where both networks and humans were mostly not able to correctly label the object.

recognize objects in unusual poses in the brain is impaired when interrupted by a mask after 40 ms viewing time, and starts taking place in a time frame of approximately 150 ms.

**Patterns of errors differ between humans and deep networks.** We next subdivided the unusual poses in two sets, the poses that EfficientNet predicted correctly (rotated-correct) vs. incorrectly (rotated-incorrect), and analysed the errors of humans and networks on these sets (Figure 4a). We found that errors made by EfficientNet transfer well to the 5 networks tested ($80.0\% \pm 7.4$ accuracy on rotated-correct condition vs. $54.2 \% \pm 5.3$ on rotated-incorrect condition, a significant difference: $t(4) = 6.99$, $p = 0.0022$)—despite the diversity of architectures and training procedures—but not to humans (difference in average performance between rotated-correct and rotated-incorrect non-significant). To investigate further whether networks and humans make similar errors (i.e., whether they classify incorrectly the same images), we computed error consistency (defined in *Statistical tests* 2.4) across networks, across humans, and between networks and humans. When computing the error consistency across networks and between humans and networks, we found that networks do similar mistakes to each other, but that human errors are on average not consistent with network errors (Figure 4b and c). Moreover, there is a greater diversity in the patterns of errors made by humans than in the pattern of errors made by networks. Overall, this suggests that the behavior of time-limited humans is different that the one of networks.

**We then sought to understand what qualitatively explained the differences in the error patterns between time-limited humans (40 ms) and deep networks.** We found that—even under acute time limitation—humans rely on different strategies than networks for recognition, relying more on object structure than their network counterparts (which rely on object details). In Figure 5a, we show examples where both networks and humans performed well. In these examples, both the structure and details of the objects are well informative about the identity of the object. In Figure 5b, we show examples of errors that networks made but humans typically didn't. A striking example is the image of a stretcher, which networks mislabeled as a matchstick. We speculate that the similarity between the stretcher's legs and matchsticks is the cause for this misclassification. Despite the resemblance to matchsticks, humans however correctly identified the object because the overall structure of the object rules out the possibility of matchsticks. Thus, we can reason that in that case, humans tended to perceive the overall structure rather than fixating on minute details, unlike networks which based their prediction on such details. In Figure 5c, we show examples where identification of

an object is challenging for humans but not for networks. For instance, images of jellyfish posed difficulties for human observers. The shortness of the exposure time, 40 ms, could potentially limit humans to recognising only the overall shape and color of the object, compatible with both a lampshade and a jellyfish. In contrast, networks are able to process fine details, such as the patterns on the jellyfish's umbrella, allowing for accurate identification regardless of pose. Figure 5d presents examples that prove to be challenging for both humans and networks. It can be noted that in all these examples, both the structure and details contain only limited information about the object identity. The impairment of both networks and humans on these images is thus consistent with our hypothesis that humans rely on structure and networks on details for recognition. These qualitative observations reinforce our conclusion that time-limited humans are not well-modelled by feed-forward networks. More examples of time-limited human failures are shown in App. B.

**Next, we tested seven large vision-language models (VLMs): GPT-4o, GPT-4-vision-preview, Claude 3.5 Sonnet, Claude 3 Opus, Gemini 1.5 Flash, Gemini 1.5 Pro and SigLIP.** With upright poses, all 6 models performed well (Gemini 1.5 Flash, GPT-4o and GPT-4-vision-preview achieving 100% accuracy, Gemini 1.5 Pro and SigLIP 98.0%, Claude 3.5 Sonnet 94.1% and Claude 3 Opus 92.2.%). However, there were significant differences in performance when recognizing rotated poses (Figure 3). Claude 3 Opus performed below the 50% chance level, achieving only 45.8% accuracy. Claude 3.5 Sonnet's accuracy was on par with pure vision networks with 62.5% accuracy. GPT-4o had an accuracy of 84.8%, SigLIP 84.0% and GPT-4-vision-preview 79.7%, performing substantially better than our best performing pure vision model, SWAG (70.1%). Remarkably, Gemini 1.5 Pro achieved an accuracy of 96.6% and Gemini 1.5 Flash an accuracy of 97.7.%, performances on par with unlimited humans (98.9%). Separating the rotated condition into two categories—rotated-correct (images correctly labelled by EfficientNet) and rotated-incorrect (images incorrectly labelled by EfficientNet)—revealed that the 7 VLMs exhibited a performance drop between these conditions, similarly to the five pure vision networks (Figure 4a). Performance differences of GPT-4o (18%), GPT-4-vision-preview (17%) and Claude 3.5 Sonnet (32%) were the most similar to the 5 pure vision networks, whose average performance gap was 26%. For Gemini models and Claude 3 Opus, the performance differences were much smaller: 4.7% for Gemini 1.5 Flash, 2.7% for Gemini 1.5 Pro and 2.4% for Claude 3 Opus. For both unlimited and time-limited humans, the performance difference was not significant between the two conditions. This suggests that errors from EfficentNet transfer better to most VLMs than to humans.

## 4 Discussion

Our results demonstrate that the human visual system is more robust than deep networks for vision and than most vision-language models at recognizing objects in unusual poses. Indeed, with unlimited viewing time, human performance is not affected by unusual poses, whereas deep networks are fooled by them. Only in the very limited viewing time condition (40 ms) does the performance of humans deteriorate to the level of deep networks.

### 4.1 The gap still exists

Our finding is in contrast to a recent study (Geirhos et al., 2021) which showed that deep networks are closing the performance gap with humans on many out-of-distribution visual tasks. Our study differs from this recent study in two major ways: (1) We are considering a transformation (object pose) which affects the global structure of the image, unlike the distortions studied in (Geirhos et al., 2021) which mostly affected the local texture of images (e.g., blur, color modifications, stylization); (2) Geirhos et al. (2021) compare network performance to time-limited humans only. The human subjects in their study were given 200 ms to view the image, followed by a mask. We find that viewing time does affect the robustness of recognition in a crucial way. When time-limited to 40 ms or 150 ms (followed by a mask), the performance of humans substantially degrades at recognizing objects in unusual poses.
Regarding the deep networks, it is notable that the architecture (e.g., visual transformers vs. convolutional neural networks), loss function (e.g., self-supervised vs. supervised), training set size (e.g., 3.6 billion images for SWAG) and modality (pure images, or images and text) did not affect their performance on the task (with the exception of Gemini, see discussion below). This observation adds to a literature showing that scaling alone seems insufficient to match human internal representations and performance in visual tasks

(e.g., (Shankar et al., 2020; Fel et al., 2022; Muttenthaler et al., 2022; Linsley et al., 2023a;b)). We also note that networks do not benefit from additional time, as their processing time is fixed by the nature of their feed-forward architecture.

## 4.2 Main limitation of the current study

A critical extension of this study would be to systematically explore how viewpoint affects the performance of humans and networks. In Abbas & Deny (2023), the effects of different rotation axes (image-plane rotation vs. 3D rotation) and rotation range are investigated in detail for networks. It is found that networks that have been trained with image rotation as a data augmentation scheme perform well on image-plane rotations, but are still impaired on 3D out-of-plane rotations. It is also found that networks perform at their worst on objects rotated 90° from their canonical pose, regardless of rotation axis. It would be interesting to study whether time-limited humans have the same or different failure modes, as this would shed light on their respective mechanisms.

While our current study doesn't allow a systematic exploration of these factors, we can still comment on some of the typical failure modes of time-limited humans. It appears that many challenging images are such that the major axis of the object is occluded due to out-of-plane rotation, causing one end of the major axis to point towards the observer (Marr & Nishihara, 1978). Examples include the corkscrew and the jellyfish in the rotated correct image (Figure 7) as well as the violin and the crib in the rotated incorrect image (Figure 8). In other cases, it seems that occlusion of a signature detail of an object, such as the watch face or a lawnmower motor, also hinders human recognition performance. This also happens most frequently due to out-of-plane rotations. It is important to notice, however, that there are images that are challenging despite only involving in-plane rotation, such as the jellyfish and the parachute in the rotated incorrect image. Thus, a challenging rotation condition is often quite object specific.

For human participants, it is difficult to obtain quantitative results of recognition as function of rotation for real objects, as each individual object cannot be presented to each observer in different rotation conditions (as there is heavy transfer of recognition across rotations). Different objects, on the other hand, will be differently affected by the same rotations. Artificial objects may be better stimuli for the quantitative characterization of the effect of rotation on human and network recognition ability, as have recently proposed (Bonnen et al., 2024).

## 4.3 Why do humans need the extra time?

Our results are in line with a tradition of psychology studies (Shepard & Metzler, 1971; Jolicoeur, 1985; Edelman & Bülthoff, 1992; Kosslyn et al., 1994; Sofer et al., 2015; Jeurissen et al., 2016; Kallmayer et al., 2023; Mayo et al., 2024) showing that humans need more time to accomplish more challenging visual tasks. Closest to our work, Jolicoeur (1985) showed that the naming time of sketches of everyday objects was proportional to the difference in orientation with their upright pose. Below, we list some of the possible mental processes that could be responsible for this extra time:

**1) Evidence accumulation:** The increased time could be needed for evidence accumulation. For example, it is known that different types of information about a visual stimulus (e.g., coarse vs. fine features, low contrast vs. high contrast features) propagate to cortical visual areas with different timings (Van Rullen & Thorpe, 2001; VanRullen & Thorpe, 2002). It might thus be necessary to see the stimulus for a certain time for all relevant information to arrive to cortex. Additionally, stimulus information always involves a certain amount of noise and uncertainty, which can be filtered with increased viewing time (Olds & Engel, 1998; Perrett et al., 1998).

**2) Recurrence within the visual system:** The visual system is known to be highly recurrent (Felleman & Van Essen, 1991; Suzuki et al., 2023). Previous evidence suggests that visual information is mainly processed in a feed-forward way during the first 80 ms after exposure (Liu et al., 2009; Wyatte et al., 2014; Cichy et al., 2016; Mohsenzadeh et al., 2018), after which recurrent processes start taking place within the

visual system (Liu et al., 2009; Thorpe, 2009; Wyatte et al., 2014; Cichy et al., 2016; Kar & DiCarlo, 2021), e.g., from extrastriate areas to V1. Moreover, other studies have suggested that backward masks, such as the ones used in this study, disrupt these recurrent processes (Lamme & Roelfsema, 2000; Lamme et al., 2002; Breitmeyer & Ogmen, 2006; Fahrenfort et al., 2007; Macknik & Martinez-Conde, 2007). The observed drop in human performance from 150 ms to 40 ms viewing time could thus be explained by these recurrent processes taking place in the 150 ms condition but being interrupted by backward masking in the 40 ms condition. From a computational standpoint, there is mounting evidence that these recurrent circuits provide important functions such as grouping and filling-in of features, and are important for object identification in noisy real-world situations, and particularly when objects are partially occluded (Wyatte et al., 2014). It is an interesting possibility that incorporating such recurrence in deep networks could bring human-like robustness to recognition of objects in unusual poses. Some efforts to incorporate recurrence in deep learning models of the visual system exist (Wyatte et al., 2012; O'Reilly et al., 2013; Spoerer et al., 2017; Tang et al., 2018; Kietzmann et al., 2019; Rajaei et al., 2019; Nayebi et al., 2022; Goetschalckx et al., 2023), but to our knowledge not in the context of recognizing objects in unusual poses.

We note that evidence accumulation and recurrence are not mutually exclusive and, in fact, recurrent circuits are one possible neural implementation to allow evidence accumulation.

**3) Recurrence between the visual system and other systems:** The neural substrate is also highly recurrent between the visual system and other brain regions (Felleman & Van Essen, 1991), such as the perirhinal cortex in the medial temporal lobe (Bonnen et al., 2023) and many frontoparietal areas, such as the prefrontal cortex (PFC) and the Frontal Eye Fields (FEF). Feedback processes originating in frontoparietal areas through reciprocal connections to striate cortex provide attentional support to salient or behaviorally-relevant features (Wyatte et al., 2014). FEF, in turn, play a crucial role in the initiation and execution of saccadic eye movements. The additional processing time could be needed to perform saccades across the object to encode the objects' subparts with successive fixations. Medial temporal cortex (MTC) has been recently shown to support object perception by integrating over multiple glances or saccades (Bonnen et al., 2023). However, these recurrent interactions between the visual systems and other areas are thought to take place at least 150 ms after the first glance (Wyatte et al., 2014; Bonnen et al., 2023). They can only partly explain the performance for unusual poses we observe, since we already see a great improvement from 40 ms to 150 ms, before these interactions could take place.

### 4.4 Time-limited humans are not feed-forward deep networks

Both time-limited humans and networks have similar error rates at recognizing objects in unusual poses. This result could suggest that time-limited humans are well-modelled by feed-forward deep networks, as proposed by (Yamins et al., 2014; Rajalingham et al., 2018; Kar et al., 2019; Serre, 2019; Dapello et al., 2020; Kar & DiCarlo, 2021; Dapello et al., 2022; Muttenthaler et al., 2022; Bonnen et al., 2023; Dehghani et al., 2023). Here, our analysis of the error patterns of time-limited humans vs. networks suggests otherwise. While the performance of networks and time-limited humans is similar on unusual poses, there is little consistency in their patterns of error. Networks share consistent patterns of error with each other, which are inconsistent with the patterns of error of humans. In addition, humans are overall more unpredictable in their error patterns than networks. Moreover, a qualitative analysis of the problematic images for networks vs. humans suggests that networks make errors when they overlook the overall structure of the object and rely instead on misleading details or textures. Conversely, time-limited humans tend to make errors when the overall structure of the object is not clearly disambiguating the object identity, and where one should instead rely on details of the object and textures to make the correct decision. This discrepancy is in line with previous studies (Geirhos et al., 2018; Rajalingham et al., 2018; Geirhos et al., 2021; Wichmann & Geirhos, 2023) which have shown that deep networks do not make the same mistakes as time-limited humans. In particular, Geirhos et al. (2018) showed that deep networks are prone to a bias towards texture, whereas humans are prone to a bias towards shape of objects in their perceptual decisions. Our results suggest that this discrepancy persists under extreme time limitation, and that feed-forward deep networks are not adequate models of time-limited humans. However, to fully disentangle the effect of texture vs. structure, one would need to design an experiment where the texture of 3D objects is manipulated independently from their structure.

### 4.5 On the role of retrospective thinking in humans and vision-language models

In the current study, humans were presented with an image followed by a two-alternative-forced-choice question. This provides humans with an opportunity to utilize retrospective thinking. For instance, a subject who initially failed to identify an image of a rotated stretcher, could reconsider their interpretation after seeing the answer alternatives 'stretcher' and 'matchstick'. Deep networks for pure vision, on the other hand, are not capable of such retrospective thinking; they classify the object based on the predefined 1000 classes without the possibility of a retrospective thought process when given the two possible answers. Conversely, large vision-language models are given the opportunity to ponder the content of the image after being fed the two labels, like humans. Their performance is notably better than deep networks for pure vision (except Claude), and even matching human performance in the case of Gemini. The impressive performance of vision-language models suggests that they might be able to exploit similar mechanisms for image recognition as humans, which could for instance involve retro-fitting the labels to the image. However, an alternative explanation is that they could have been fed a different data diet involving more images, or more rotated objects than their pure vision counterparts. Indeed, Abbas & Deny (2023) found that increasing the training dataset size generally resulted in better performance in rotated object recognition. The exceptional performance of Gemini could even be due data contamination issues, since the objects we selected were available online. Because of the limited documentation available for these models, it is difficult to come to a firm conclusion regarding the mechanisms underlying their abilities. Finally, the current study suggests that the human visual system improves its accuracy by using more time, possibly by means of recurrent processing. It is a highly promising hypothesis that AI systems for vision may benefit from such recurrent processes, if we can understand their principle in the brain.

## Acknowledgements

This work has been supported by an Aalto Brain Center (ABC) grant to N.O., and an Academy of Finland (AoF) grant to S.D. under the AoF Project: 3357590.

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

## Appendix

All code and data is available at `https://github.com/BRAIN-Aalto/unusual_poses`.

## A    Testing Large Vision-Language Models

We tested 6 large vision-language models (VLMs), Gemini 1.5 Flash, Gemini 1.5 Pro, GPT-4o, GPT-4-vision-preview, Claude 3.5 Sonnet and Claude 3 Opus, via their API. Figure 6 shows an example of the used prompt and models' answers.

Swapping answers A and B did not change the accuracy of the VLMs. Indeed, we ran all VLM analyses also with switched choices. That did not have any effect.

Changing the prompt did affect the results, but we carefully selected the prompt that worked well for all VLMs.

For example, we also tried the following prompt with the rotated-incorrect images:

*The image represents either [label1] or [label2]. Which is it? Answer in one or two words.*

For GPT-4o, this prompt led to 78% correct answers, whereas our original prompt had 76% correct answers. However, for Gemini, this prompt led to 91% correct answers, against 95% for our original prompt. This illustrates the variability that comes with changing the prompt.

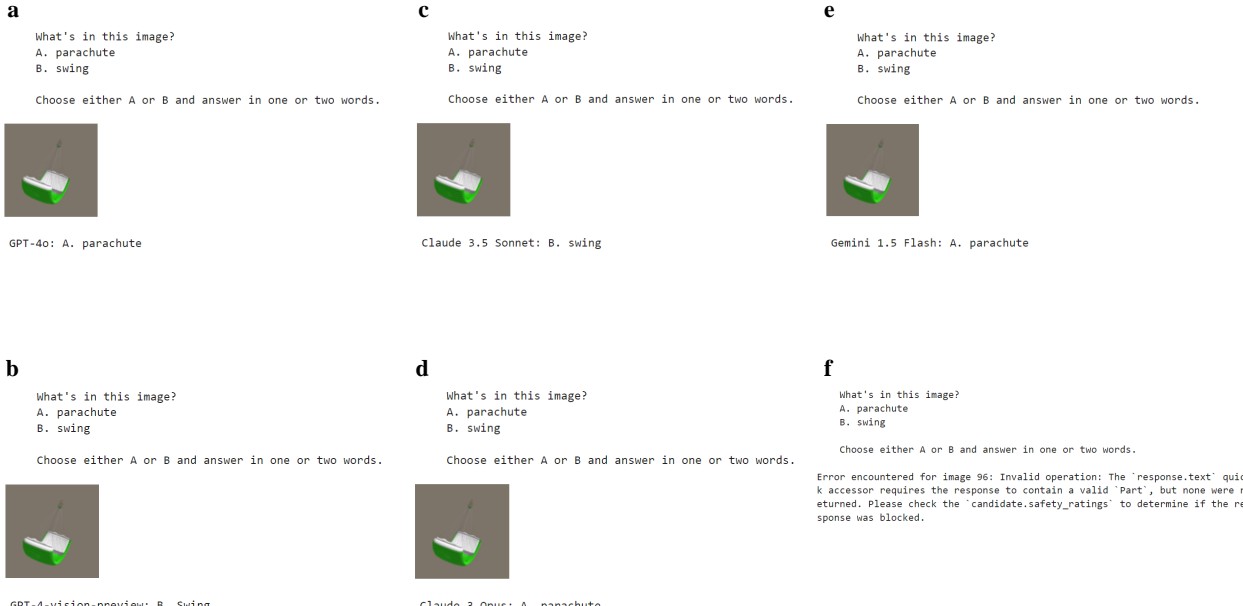

Figure 6: **Examples of a prompt and models' answers in API mode. a)** GPT-4o, **b)** GPT-4-vision-preview, **c)** Claude 3.5 Sonnet, **d)** Claude 3 Opus, **e)** Gemini 1.5 Flash and **f)** Gemini 1.5 Pro (example of safety-blocked instance, which was disregarded in our analysis).

## B    More examples of time-limited (40 ms) human failures

Figure 7 and Figure 8 show more examples of images that were typically challenging for time-limited humans, respectively in the rotated-correct and rotated-incorrect conditions.

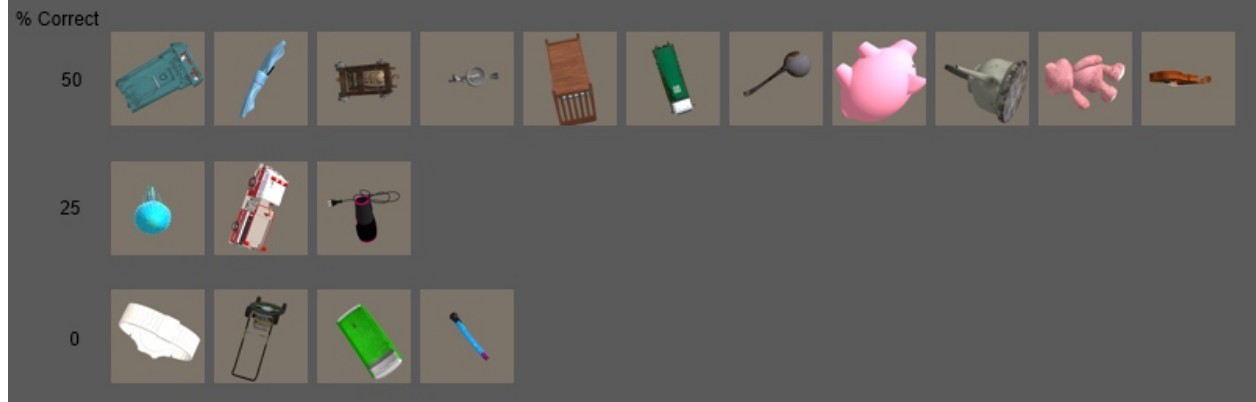

Figure 7: **Examples of images where time-limited humans were mostly incorrect, for the rotated-correct condition.** First row: humans were <50% correct; Second row: <25%, Third row: 0%

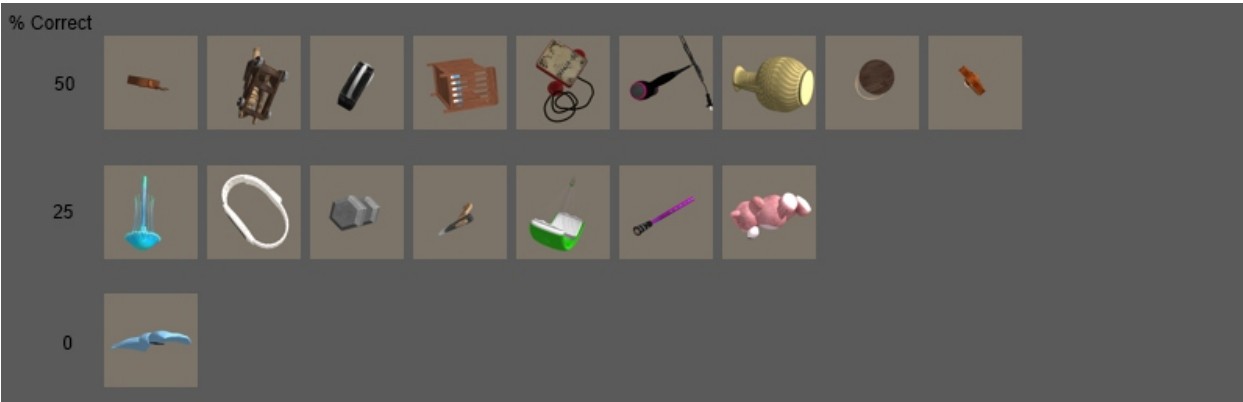

Figure 8: **Examples of images where time-limited humans were mostly incorrect, for the rotated-incorrect condition.** First row: humans were <50% correct; Second row: <25%, Third row: 0%

## C  Deep Networks (pure vision) details

### C.1  Dataset and Training Objectives

We used six different models in this experiment: Noisy Student EfficientNet-L2, SWAG-RegNetY, ViT-L16, BEiT-L/16, ConvNeXt-XL, and SWIN-L. The models were chosen for their demonstrated performance at recognizing objects in unusual poses in a prior study conducted by Abbas & Deny (2023). The models were trained on datasets of varying sizes ranging from 1 million (ImageNet) to 3.6 billion images (IG-3.6B). These models had a variety of architectures (e.g., convolutional architecture, visual transformer) and were trained under a variety of training objectives:

- Supervised learning: Visual Transformers (Dosovitskiy et al., 2020; Liu et al., 2021) and convolutional architectures (Liu et al., 2022).

- Self-supervised learning: BEiT (Bao et al., 2021).

- Semi-supervised learning: Noisy Student (Xie et al., 2020).

- Semi-weakly supervised learning: SWAG (Singh et al., 2022).

### C.2 Model Descriptions

This section describes the networks used in this study in detail.

**Standard Vision Transformer:** Standard Vision Transformer ViT-L/16 (Dosovitskiy et al., 2020) which was pretrained on ImageNet(1M) with input size 224x224.

**BEiT:** BEiT-L/16 is a Vision Transformer trained using a self-supervised training method introduced with the model (Bao et al., 2021). The method is known as Masked Image Modeling (MIM) and it is inspired by Masked Language Modeling (MLM) (Devlin et al., 2018). The model is pretrained with the self-supervised MIM task on ImageNet21k(14M) and fine-tuned on ImageNet. Even with smaller pretraining datasets, BEiT outperforms standard Visual Transformers.

**ConvNext:** ConvNeXt-XL (Liu et al., 2022), is a pure convolutional architecture which layers are designed carefully by choosing an optimal collection of architecture hyperparameters. The model undergoes training methods established for Visual Transformers, and they are known to enhance ConvNext's performance as well. The model is trained on ImageNet21k(14M).

**SWIN Transformer:** SWIN-L Transformer (Liu et al., 2021) is a hierarchical Vision Transformer. The self-attention layer of the original Vision Transformer is replaced by a Shifted Window Self-attention layer (SWSA). The SWSA layer divides the image into windows, calculating self-attention within each window. This process significantly reduces the computational time of the attention layer, making it linear with the input size instead of quadratic. Additionally, it enhances performance compared to the original Visual Transformers. The model is trained on ImageNet21K(14M).

**Noisy Student:** Noisy Student EfficientNet-L2 is pretrained using the Noisy Student paradigm (Xie et al., 2020). The model is trained on ImageNet dataset and on the noisily labeled JFT-300M dataset. It has an input size of 475x475. The model is pretrained with the data augmentation method RandAugment (Cubuk et al., 2020), which among other distortions produces rotated images of the training set.

**SWAG:** SWAG-RegNetY (Singh et al., 2022) is pretrained using a semi-weakly supervised pretraining procedure. The pretraining dataset consisted of more than 3.6 billion Instagram images, which were labeled with hashtags of 27K classes.

### C.3 Model Sources

The checkpoints for all the models we use are in PyTorch format. The models are from three different main sources: Pytorch Image Model library (timm) (ViT-L16, ConvNext-XL, SWIN-L), Hugging Face Transformers library (BEiT-L16), and Torch Hub (EffN-L2-NS, SWAG-RegNetY).

## D The 3D objects

Below are the 3D objects used in our experiments (Figures 9, 10, 11, 12, 13, 14, 15, 16, 17, 18, 19, 20, 21, 22, 23, 24, 25, 26, 27, 28, 29, 30, 31, 32, 33, 34, 35, 36, 37, 38, 39, 40, 41, 42, 43, 44, 45, 46, 47, 48, 49, 50, 51, 52, 53, 54, 55, 56, 57, 58, and 59). They are listed with the links to Sketchfab. Images of each object are also shown with the corresponding wrong labels (first upright, second rotated and correctly labelled by EfficientNet, and third rotated and incorrectly labelled by EfficientNet).

- Acoustic guitar: link

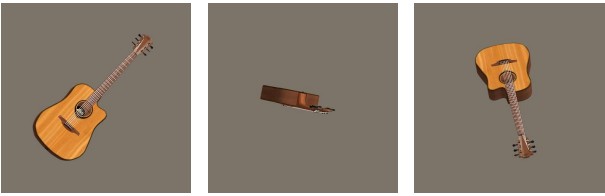

Figure 9: 1. Banjo, 2. Plectrum, 3. Whistle

- Airliner: link

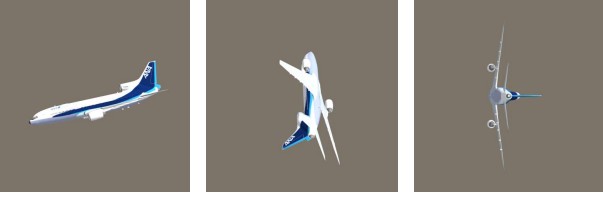

Figure 10: 1. Warplane, 2. Space shuttle, 3. Missile

- Backpack: link

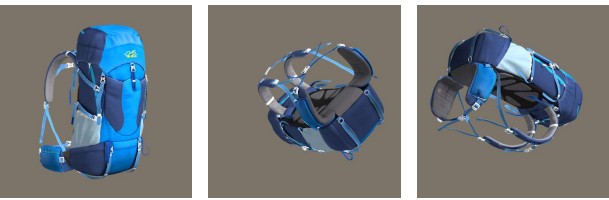

Figure 11: 1. Sleeping bag, 2. Neck brace, 3. Neck brace

- Binoculars: link

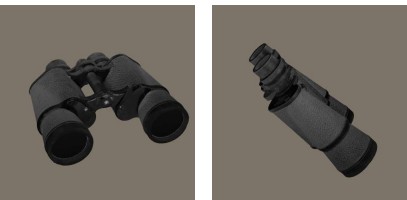

Figure 12: 1. Reflex camera, 2. Tripod

- Bow tie: object deleted

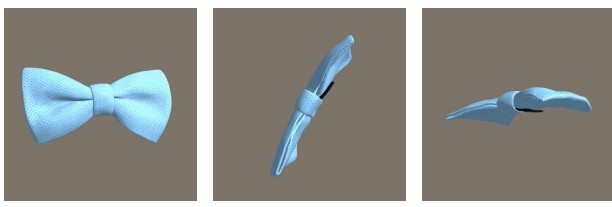

Figure 13: 1. Windsor tie, 2. Band Aid, 3. Hammerhead shark

- Cannon: link



Figure 14: 1. Rifle, 2. Hourglass, 3. Guillotine

- Canoe: link

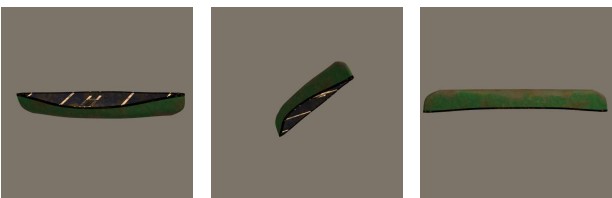

Figure 15: 1. Speedboat, 2. Hair slide, 3. Toaster

- Combination lock: link

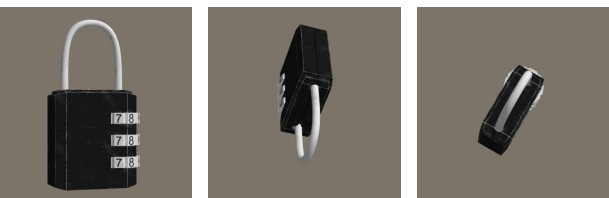

Figure 16: 1. Safe, 2. Hook, 3. Airship

- Corkscrew: link

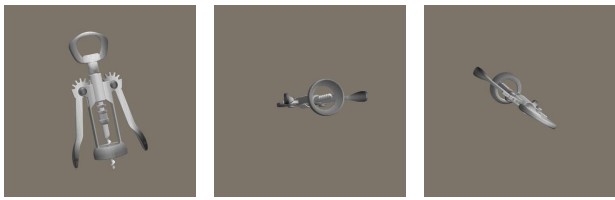

Figure 17: 1. Can opener, 2. Syringe, 3. Unicycle

- Crib: link

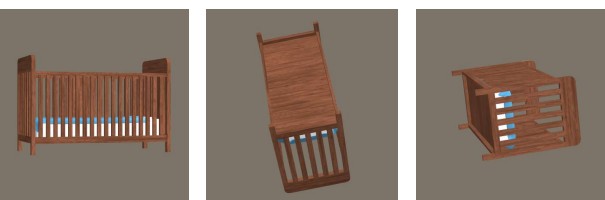

Figure 18: 1. Studio couch, 2. Plate rack, 3. Crate

- Dial telephone: link

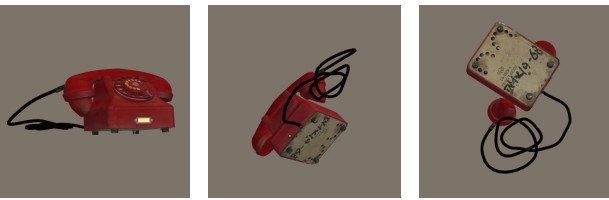

Figure 19: 1. Smoothing iron, 2. Hook, 3. Joystick

- Digital watch: link

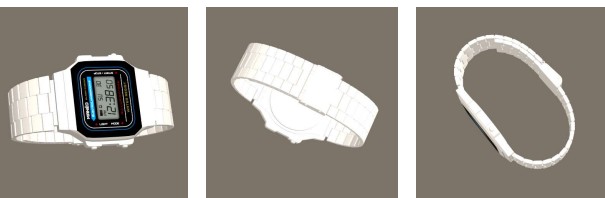

Figure 20: 1. Magnetic compass, 2. Buckle, 3. Chain

- Dumbbell: link

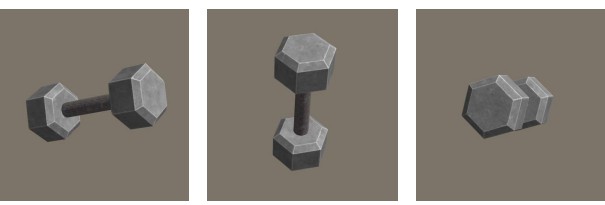

Figure 21: 1. Honeycomb, 2. Nail, 3. Honeycomb

- Electric fan: link

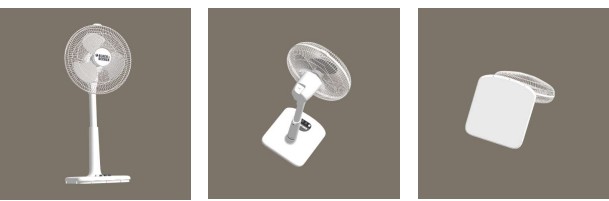

Figure 22: 1. Spotlight, 2. Microphone, 3. Strainer

- Frying Pan: link



Figure 23: 1. Spatula, 2. Gong, 3. Shield

- Fire engine: object deleted

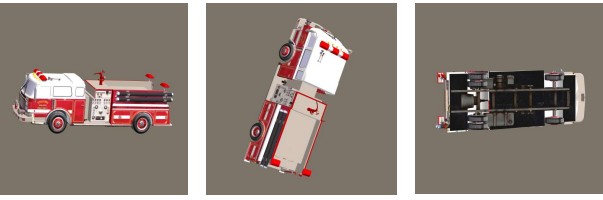

Figure 24: 1. Tow truck, 2. Trailer truck, 3. Chain

- Folding chair: link

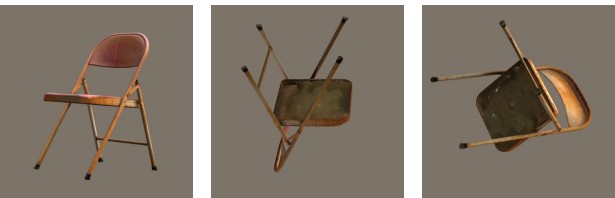

Figure 25: 1. Rocking chair, 2. Tray, 3. Shovel

- Football helmet: link

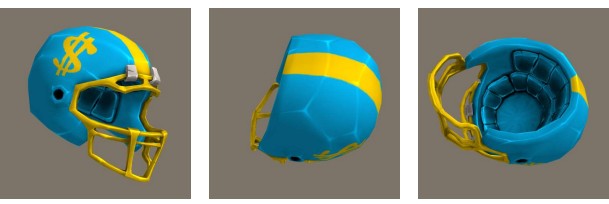

Figure 26: 1. Mask, 2. Balloon, 3. Bucket

- Forklift: link

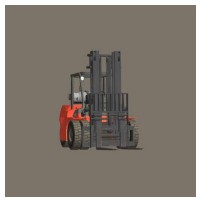 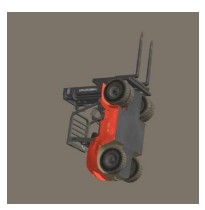 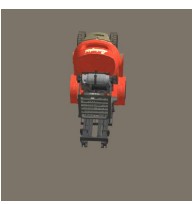

Figure 27: 1. Crate, 2. Crane, 3. Lawn mower

- Garbage truck: object disabled

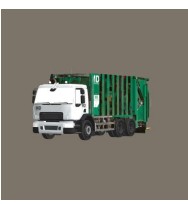 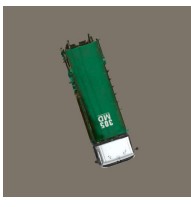 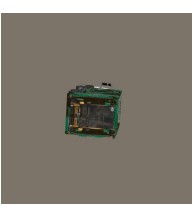

Figure 28: 1. Trailer truck, 2. Stretcher, 3. Racer

- Gasmask: link

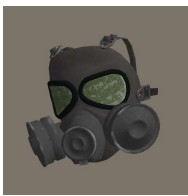 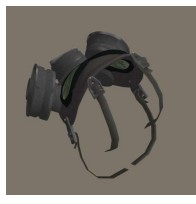 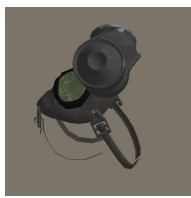

Figure 29: 1. Ski mask, 2. Binoculars, 3. Spotlight

- Hair dryer: link

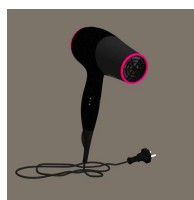 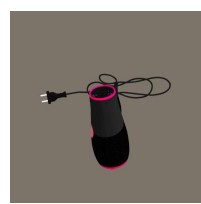 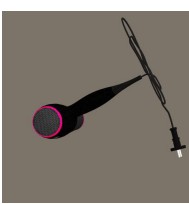

Figure 30: 1. Microphone, 2. Computer mouse, 3. Microphone

- Hammer: object deleted

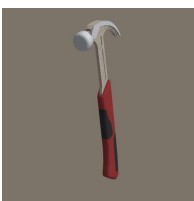 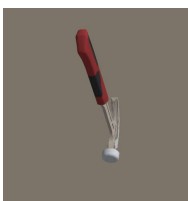 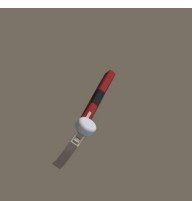

Figure 31: 1. Hatchet, 2. Can opener, 3. Bow

- Hatchet: link

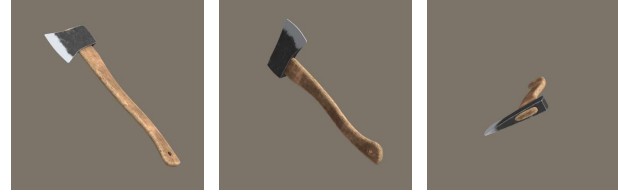

Figure 32: 1. Hammer, 2. Cleaver, 3. Carpenter's plane

- Jellyfish: link

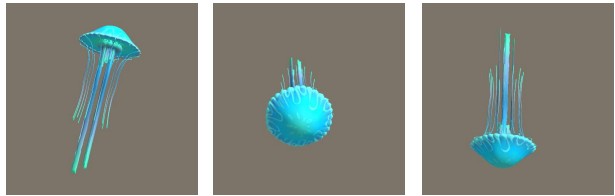

Figure 33: 1. Parachute, 2. Lampshade, 3. Lampshade

- Ladle: link

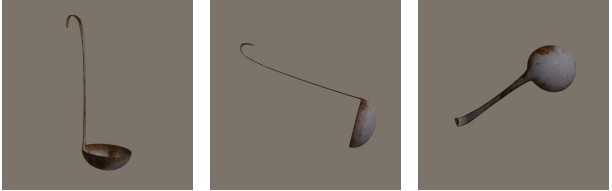

Figure 34: 1. Wooden spoon, 2. Maraca, 3. Safety pin

- Lawn mower: link

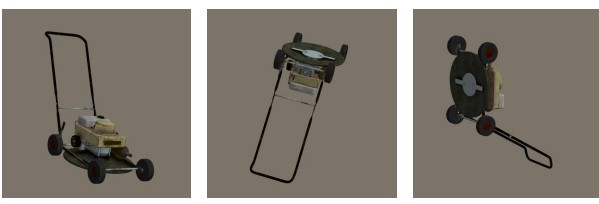

Figure 35: 1. Harvester, 2. Mousetrap, 3. Projector

- Mountain bike: link



Figure 36: 1. Tricycle, 2. Tricycle, 3. Unicycle

- Mountain tent: link

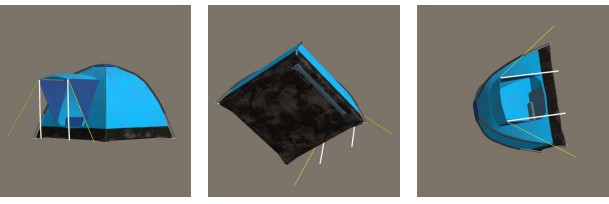

Figure 37: 1. Dome, 2. Sleeping bag, 3. Crash helmet

- Parachute: link

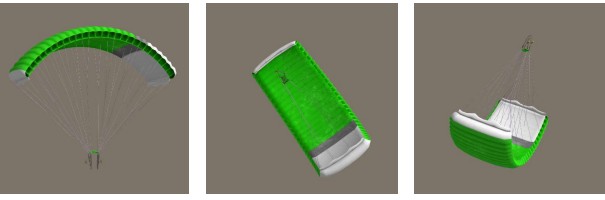

Figure 38: 1. Balloon, 2. Lighter, 3. Swing

- Park bench: link

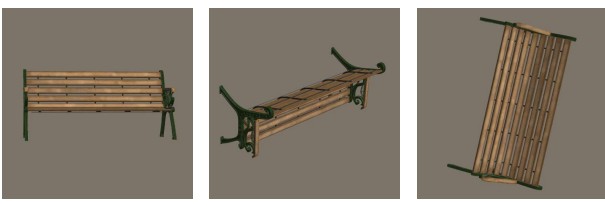

Figure 39: 1. Studio couch, 2. Paper towel, 3. Swing

- Piggy bank: link

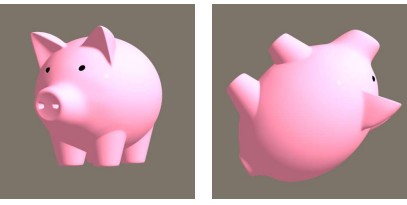

Figure 40: 1. Hog, 2. Teapot

- Pitcher: link

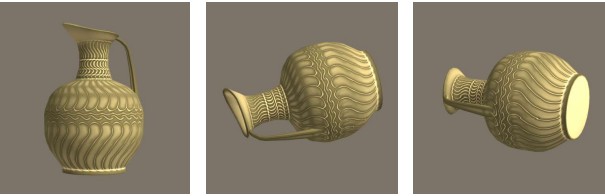

Figure 41: 1. Vase, 2. Conch, 3. Drum

- Power drill: link

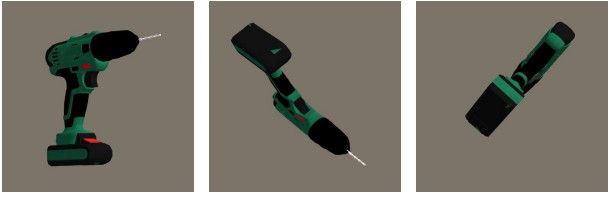

Figure 42: 1. Hair dryer, 2. Screwdriver, 3. Mop

- Racket: link

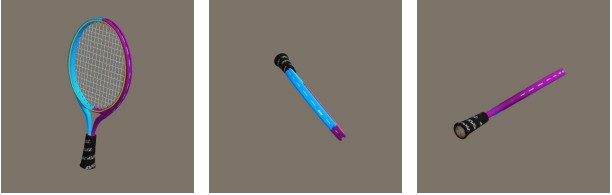

Figure 43: 1. Strainer, 2. Ballpoint pen, 3. Mop

- Rocking chair: link

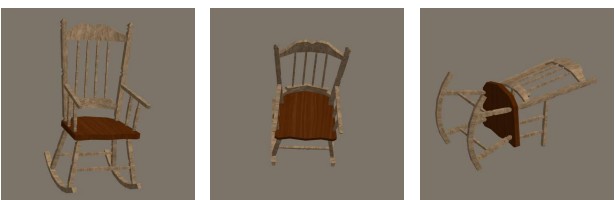

Figure 44: 1. Folding chair, 2. Throne, 3. Cradle

- Running shoe: object deleted

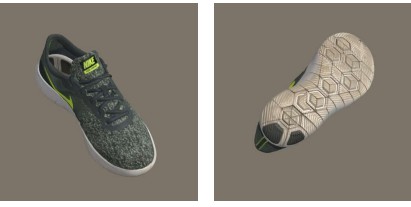

Figure 45: 1. Clog, 2. Honeycomb

- Scooter: link

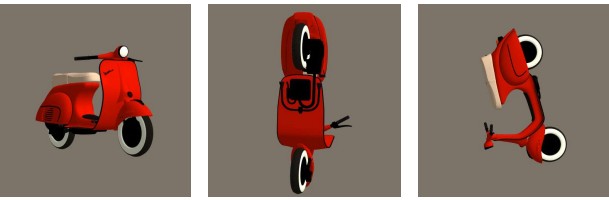

Figure 46: 1. Moped, 2. Moped, 3. Tricycle

- Shopping cart: link

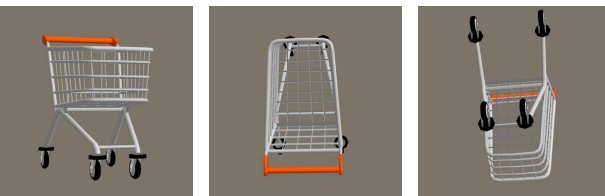

Figure 47: 1. Shopping basket, 2. Shopping basket, 3. Shopping basket

- Stretcher: link

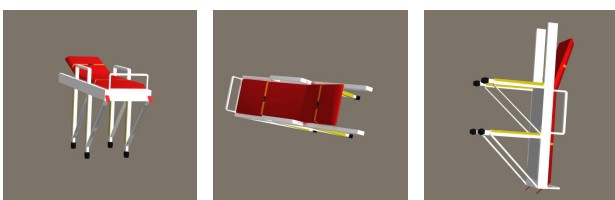

Figure 48: 1. Crutch, 2. Bobsled, 3. Matchstick

- Table lamp: link

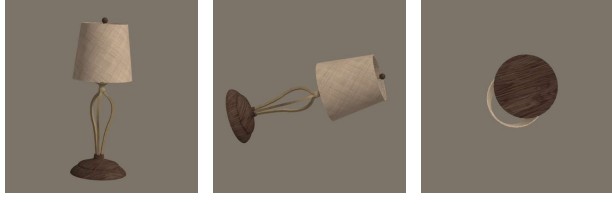

Figure 49: 1. Pedestal, 2. Plunger, 3. Toilet seat

- Tank: link

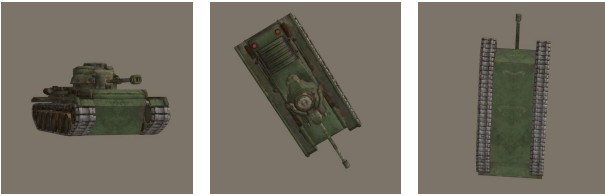

Figure 50: 1. Cannon, 2. Stretcher, 3. Pedestal

- Teapot: link

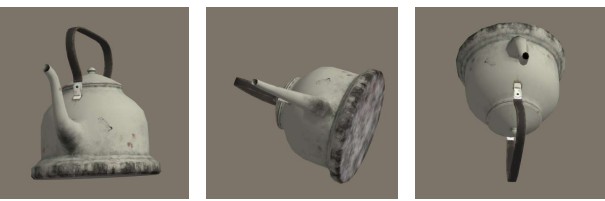

Figure 51: 1. Water jug, 2. Plunger, 3. Speaker

- Teddy bear: link

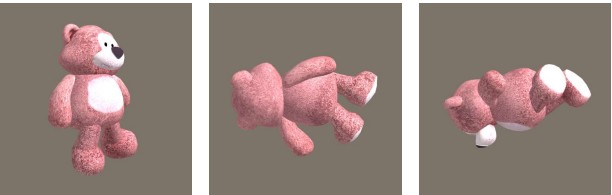

Figure 52: 1. Piggy bank, 2. Knot, 3. Piggy bank

- Tractor: link

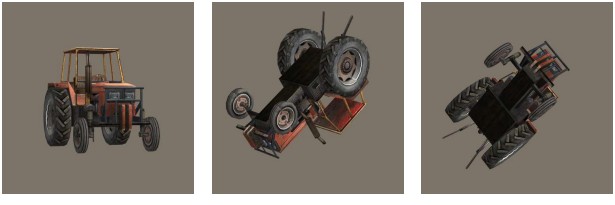

Figure 53: 1. Forklift, 2. Mousetrap, 3. Forklift

- Traffic light: link

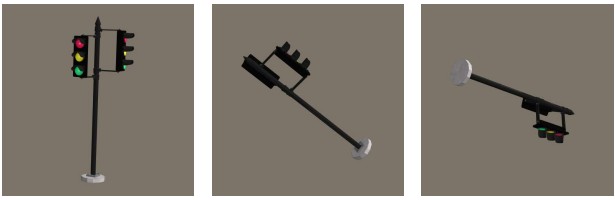

Figure 54: 1. Pole, 2. Missile, 3. Nail

- Trashcan: link

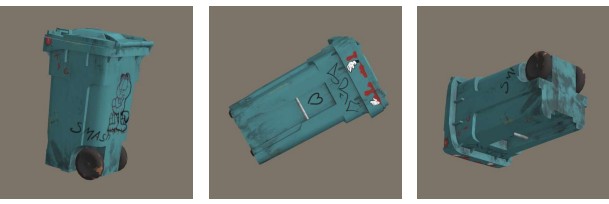

Figure 55: 1. Bucket, 2. Lighter, 3. Revolver

- Umbrella: link

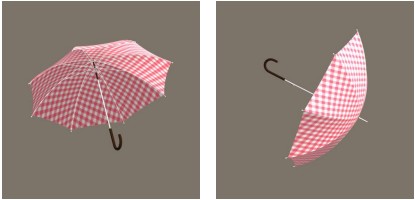

Figure 56: 1. Mountain tent, 2. Kimono

- Vacuum cleaner: link

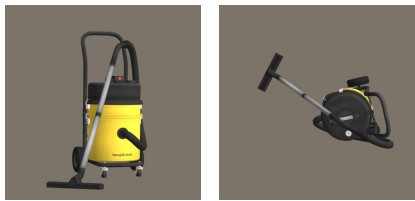

Figure 57: 1. Lawn mower, 2. Chainsaw

- Violin: link

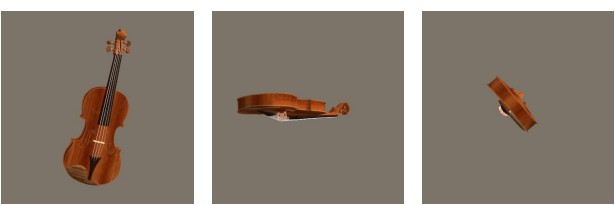

Figure 58: 1. Banjo, 2. Harp, 3. Spindle

- Wheelbarrow: link

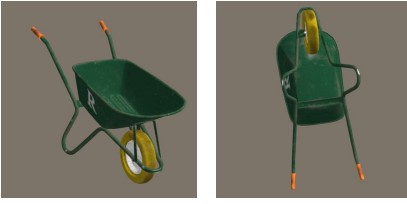

Figure 59: 1. Shovel, 2. Shovel

