# OpenReview forum: "A comparison between humans and AI at recognizing objects in unusual poses"
_TMLR — Accepted by TMLR_

### Review · Reviewer_DiFG · 2024-11-08

**Summary Of Contributions:**

This paper is a study on the robustness of various AI models, as well as humans, on the problem of recognizing objects in unusual poses in images. The authors collected a dataset of images with objects with various rotation angles on white backgrounds. They evaluated various popular purely vision based and vision-language models and performed the same evaluation with humans on various settings where humans can see the image a given amount of time. The paper finally compares performance across models and settings, patterns of errors, strengths and weaknesses and draws several conclusions such as the fact that humans and machines do not make the same mistakes.

**Audience:**

Yes

**Broader Impact Concerns:**

This study involves human experiments and data. A broader impact statement could be incorporated to describe how privacy of the participants was preserved.

**Claims And Evidence:**

Yes

**Requested Changes:**

- Incorporate a discussion on how to improve current AI systems to close the gap with humans.

**Strengths And Weaknesses:**

Strengths:

- The problem of recognizing objects in unusual poses is very interesting and differs from classical problems in out-of-distribution generalization in computer vision. This problem is rarely tackled although this paper highlights the fact that AI models have important limitations. This paper makes a step toward understanding this problem, with a clear and convincing experimental protocol and high quality analysis.

- The list of models evaluated is very diverse and contains most of the popular and best models that the vision community is currently using. ViT is now the most popular vision backbone and there is still a strong interest in CNN with ConvNeXt . For VLMs, the authors evaluated all the best and most popular models such as Gemini, Claude and GPT-4o.

- The paper incorporates strong knowledge in neurosciences. For example, the parameters of the human experiments, 40ms / 150ms display time, are not chosen randomly and are based on other neuroscience studies. Moreover, the paper presents interpretation and connections with various components of the brain involved in perception.

- The paper is very well presented, clear and well written. In particular Figure 3 is very clear and summarizes well the conclusions of the study.


Weaknesses:

- One important conclusion is not much discussed in the paper: VLMs exhibit much better performance than vision models. What is your intuition for that ? Do you think language is a key difference ? Maybe it is just due to the details of how to probe these models ? Also, what makes Gemini a much better model than even the other VLMs ? You give hints on the fact that it could be the training data, what is your intuition ? The gap may also come from the prompting technique. Have you tried different prompts for all the VLMs ?

- The biggest weakness of the paper is the lack of practical analysis of the results, and lack of recommendation for building better models. The paper identifies a strong limitation of AI models compared to humans, but does not elaborate on how to fix this limitation, even by giving intuitions or ideas. Now that you have these results, how would you improve existing models ?

- The study is conducted with objects on white backgrounds. In realistic scenarios, some objects appear more frequently on some backgrounds, and this prior knowledge is used to help recognition, probably accelerating the process for humans. For example, if the object is a car, it will probably appear more frequently on roads, and seeing the road for 40ms for humans could be enough to help the recognition. This setting would be more realistic. However I understand the idea behind not being sensitive to these correlations, and I appreciate that the authors discuss this point in the caveat 2 section. Another study with background could be an extension of this work.


Questions:

- For human experiments, you mention “normal vision or corrected-to-normal vision” ? Could you give more details on that ?

- The protocol for comparing error patterns is to look at errors made by EfficientNet and checking whether humans make the same mistakes. Have you considered the reverse process of starting from human errors, and checking AI models ?


Conclusion:

This paper presents really cool findings, and highlights a strong limitation of vision models. It shows that VLMs are very good at unusual pose recognition but that there still exists a gap with humans. On the weaker side I would have liked to have more recommendations or ideas on how to close this gap, but overall I think that the study in itself is enough for acceptance at TMLR, and that closing the gap could be part of future work.

---

> ### Author Response · Authors · 2024-11-15
> **Thank you for your review - answers to your specific remarks**
>
> Dear Reviewer,
>
> Thank you for your complimentary review and valuable remarks on our work. Below, we try to answer your questions to the best of our knowledge.
>
> 1.      Performance of Vision-Language Models (VLMs) vs. Vision Models
> We believe VLMs outperform traditional vision models due to a key factor discussed in Section 4.4. "On the role of retrospective thinking in humans and vision-language models". When humans see possible label choices after seeing an object, they can retrospectively consider which label best matches the object, a cognitive process that pure vision models lack. Vision models must choose from a predefined set of classes (e.g., 1,000 ImageNet labels) without the flexibility of analyzing the image in light of the two possible labels (although we do force them to choose between two labels by looking at the argmax prediction over two classes). In contrast, VLMs receive the two label choices simultaneously with the image, allowing them to “fit” the label choices to the image and choose a better answer.
> As for whether language is a crucial difference, we believe that multimodality strengthens VLMs' understanding and decision-making abilities. However, we cannot conclusively identify all factors behind Gemini’s performance, as there is limited documentation available. One possible advantage could be Gemini's exposure to more rotated objects in its training data, or even overlap with the specific objects we used, which were sourced online.
> Regarding prompting techniques, we experimented with various options, which however always included selecting between the two label choices. We found our prompting technique to be the best because when using other prompts, Claude for instance, often chose an answer outside the given choices.
> => We expand section 4.4 to answer the reviewer’s questions to the extent possible.
>
> 2.      Practical Analysis and Recommendations for Model Improvement
> Our current findings do not enable us to make direct recommendations for developing more robust models. Further research is needed into the mechanisms of the human visual system, where we hypothesize that recurrent processing plays a critical role. We propose that recognizing upright images relies primarily on feed-forward pathways, while more challenging cases, like rotated images, require recurrent processing, potentially both within and between levels in the visual hierarchy. Current deep networks lack such recurrent processing, operating solely in a feed-forward manner. In our next study, we are investigating this recurrence in the human visual system to better understand what underlies human robustness, with the hope of eventually applying these insights to model improvement.
> => We have added a short concluding remark about what we think the current models are lacking.
>
> 3.      Use of Objects without Backgrounds
> We indeed designed our dataset without backgrounds to ensure that any performance differences were based alone on rotation rather than contextual clues. We agree with the reviewer, however, that humans are likely to greatly benefit from ecologically valid context, and experiments with such context are something we definitely want to try going forward.
>
> 4.   	Normal Vision vs. Corrected-to-Normal Vision
> In our study, "normal vision" refers to observers with normal visual acuity without any corrective measures (e.g., glasses, laser surgeries). "Corrected-to-normal" vision refers to individuals who achieve normal visual acuity by using vision aids such as glasses.
> => We now add this information to the manuscript in the Methods section (p4).
>
> 5.   	AI and Human Error Patterns
> We created two conditions using images that EfficientNet either labeled correctly or incorrectly. This approach enabled us to identify challenging poses and label alternatives for both humans and networks, but is indeed partially biased against networks (as we discuss in Caveat 1). Collecting a similar dataset based on human labeling and errors would be challenging, as once humans correctly recognize an object, determining a "second-best guess" becomes difficult, and as a human has seen an object in one pose, we cannot show the object again in another pose (because the human remembers seeing that object). However, creating a dataset using human errors, and investigating whether these errors transfer to other humans and AI is an intriguing question which would be worth exploring.

---

### Review · Reviewer_ZRm5 · 2024-11-25

**Summary Of Contributions:**

This work presents a valuable comparative study of object recognition performance given challenging viewpoints. The study compares the performance of humans as well as state-of-the-art artificial recognition pipelines. This includes both "pure vision" feed-forward recognition networks, as well as large "vision-language" models.

The experimental setup largely makes sense.
- Object selection: The authors select 51 object categories from the ImageNet classes which fulfill the following criteria: (a) Objects should have a canonical upright pose (this would e.g. include chairs, cars), (b) changes in pose should be reflected in significant appearance changes (this would exclude e.g. various types of balls), and (c) should be recognisable by a reference ("pure vision") model (in this case EfficientNet-L2 with large-scale training).
- Stimulus generation: Given one 3D model per selected category, the authors then apply a random rotation relative to some canonical pose, and then render the model against a uniform background.

- Task definition: To determine object recognition performance, a two-forced choice question is put together for each stimulus. The two choices are: (a) the correct label, (b) the highest-scoring incorrect label obtained from the outputs of the aforementioned reference model (EfficientNet-L2). If the reference model correctly identifies the stimulus, this is thus the label with the second-highest softmax probability.

The authors outline two potential issues with this setup, namely that (a) the incorrect labels might be biased against artificial systems given that they are derived from one such system, and (b) that the stimuli are not realistic enough. They convincingly address these two caveats in my view.

Furthermore, they adapt this setup a little taking into account the different natures of the "experimental subjects" under consideration.

- "Pure vision" models output one probability per object class from a closed set, and thus the answer to the forced choice question is derived by comparing the probabilities of the two classes in question, even if neither is the model's first choice.

- VLMs require some kind of instruction to guide processing, so a fixed prompt template is devised in order to elicit a specific response. This prompt is as follows: "What’s in this image? A. [label 1] B. [label 2] Choose either A or B and answer in one or two words." If the VLM provides an undesirable third answer as they are sometimes wont to do, the answer is considered incorrect -- which is similar to the approach above.

- The human visual system is known to recruit recurrent processes over timescales that can vary depending on how challenging a visual task is. Artificial networks typically involve a bounded or even fixed amount of computation. Thus human subjects were presented with stimuli at one of two very short time scales: 40ms and 150ms. In the former case, this amount of time was picked as it likely precludes heavy use of recurrent processing, allowing for better comparisons with standard feed-forward vision networks. For reference, some human subjects were also given unlimited processing time as well.

The main findings:

- Humans essentially demonstrate no degradation in performance when given unlimited viewing time for recognising objects in unusual poses.
- The picture starts to change when limited time is available, especially in the 40ms case for which humans slightly underperform relative to the more advanced VLMs.
- However, a qualitative analysis of the results shows that the mistakes made by humans are of a different nature even if the results are similar quantitatively: Humans tend to make mistakes when shape cues are misleading, and texture cues would be more helpful. This accords with the results of e.g. Geirhos et al. (2018) that show how DNNs and humans appear to rely on different cues for object recognition.

**Audience:**

Yes

**Broader Impact Concerns:**

I have no ethical concerns.

**Claims And Evidence:**

Yes

**Requested Changes:**

My main concern is the following: I think a critical part of the study which I feel is underexplored is the comparison of mistakes among the different "participants" in the study. Is there perhaps a way to more systematically arrive at a characterisation of the different mistakes? Either through a lengthier/more rigorous qualitative analysis, or via a more controlled generation of objects in unusual poses?

For example with regards to the latter: The authors pick object classes with a canonical "upright" pose, but then randomly rotate the objects along either the x, y, and z axis. (For the sake of the discussion, I will assign directions to x, y, and z -- namely: up, side, and forward respectively) This equal treatment of axes I find to be a little problematic. Certain objects that have an upright position, such as chairs, can be rotated arbitrarily on the ground plane (w.r.t. to the up/x axis) without this resulting in a challenging image. In fact, such rotations will most likely occur in a natural image collection.  Similarly, a rotation around the z- or forward axis can perhaps be simulated via first rendering the 3D model to an image, and then applying an image rotation in 2D -- perhaps an augmentation operation that was used for training the more advanced image models. Significant rotations around the y- or side/lateral axis on the other hand (e.g. a chair that has been tipped over) are much less likely to occur in the training set, and thus pose a different challenge.

Aside from the choice of rotation axis, also the degree of rotation matters quite a bit, but as far as I can tell the rotation angle was sampled uniformly?

Making these kinds of distinctions is very challenging especially for many different types of object classes at once, but perhaps for a limited subset of the chosen classes it would have been worth doing some kind of quantitative analysis that attempts to put a number on how difficult a certain rotation is, or maybe to simply just break down the results by axis of rotation. This might shed more light on how humans and artificial networks differ.

Given that the stimuli are artificially generated, introducing some textural confounders (similar to the aforementioned Geirhos et al. 2018 study) might also give a more nuanced and well-grounded picture.

Further minor concerns:

- I wonder what kind of experiments the authors performed to make sure results are robust against certain reasonable variations of the setup. For example, why just pick one 3D model per object type? What about the text prompt provided to VLMs? For example, what happens if you conduct the experiment twice with the same VLM, but by consistently switching the choices of A & B, or by varying the text prompt?

- Deriving an answer to the two-forced choice question by simply looking at softmax activations and picking the bigger one largely makes sense, but should there perhaps be a cutoff in terms of how high the probability needs to be before accepting a correct answer? E.g. let's say that if for a certain stimulus, the DNN would assign to the correct label a probability of 0.01, and to the incorrect label a probability of 0.009, and to a third label entirely a probability of 0.9. Would this still count as a correct answer given that 0.01 is larger than 0.009, and the 0.9 probability answer is ignored?

**Strengths And Weaknesses:**

Overall, I believe this to be a nice and valuable study. It proposes a new experimental setting w.r.t. to other works that have probed robustness of DNNs to unusual pose changes, especially through the comparison with human subjects under various viewing conditions as well as a broad selection of state-of-the-art vision models. The experimental settings are explained and justified in detail. As described above, they mostly make sense (except for one key aspect discussed in the next section). The results are convincing and presented nicely together with a lengthy discussion that attempts to put them in a broader context with a focus on known differences between human and artificial vision systems.

I don't find the results to be surprising, but that does not detract from its value at all since it is my view that artificial recognition systems (whether "pure vision" DNNs or VLMs) are highly overrated when it comes to robust object recognition, and that relatedly papers focusing on the flaws of such systems tend to be under-appreciated. However, in light of this view, I think that one aspect of the experimental design could use some strengthening, namely the choice to uniformly sample from the space of rotations, when in fact different kinds of rotations (e.g. rotations on the ground plane for upright objects vs. other more uncommon rotations) pose very different challenges. See next section for a lengthier discussion of this point as well as further suggestions for improvements.

---

> ### Author Response · Authors · 2024-12-09
>
> Dear Reviewer,
>
> Thank you for your review and valuable comments, to which we answer below:
>
> **Main concern: exploring the role of rotation axis and especially natural/unnatural rotation axes**
>
> Thank you for raising this good point about rotation axes, some being more natural than others. Although this exact question has not been addressed in Abbas & Deny (2022),  the effects of different rotation axes (in-image-plane rotation vs 3D rotation) is investigated in detail (for networks only) in Abbas & Deny (2022), fig 5. Moreover, they investigate thoroughly other factors such as the effects of combining rotations and scaling (fig 6), as well as the effect of adding a natural background for the object (fig 13).
>
> For human participants, it is difficult obtain quantitative results of recognition as function of rotation for real objects, as each individual object cannot be presented to each observer in different rotation conditions (as there is heavy transfer of recognition across rotations). Different objects, on the other hand, will be differently affected by the same rotations (as the reviewer also pointed out). Probably, artificial objects are better stimuli for the quantitative  characterization of the effect of rotation on human recognition ability. However, to get some insight on what might cause human observers recognition challenges, we have collected the specific rotation conditions, where human performance was low (averaged over observers) in two images in Appendix B.
>
> It appears that many challenging images are such that the major axis of the object is occluded due to out-of-plane rotation (cf. Marr  & Nishihara, 1978), causing one end of the major axis to point towards the observer. Examples include the corkscrew and the jellyfish in the rotated correct image as well as the violin and the crib in the rotated incorrect image. In other cases, it seems that occlusion of a signature detail of an object, such as the watch face or a lawnmower motor, also hinders human recognition performance. This also happens most frequently due to out-of plane rotations. It is important to notice, however, that there are images, that are challenging despite only involving in-plane rotation, such as the jellyfish and the parachute in the rotated incorrect image. Thus, a challenging rotation condition is often quite object specific.
>
> *Marr, David, and Herbert Keith Nishihara. "Representation and recognition of the spatial organization of three-dimensional shapes." Proceedings of the Royal Society of London. Series B. Biological Sciences 200.1140 (1978): 269-294.*
>
> *Abbas & Deny (2022) : Progress and limitations of deep networks to recognize objects in unusual poses https://arxiv.org/abs/2207.08034*
>
> **Aside from the choice of rotation axis, also the degree of rotation matters quite a bit, but as far as I can tell the rotation angle was sampled uniformly?**
>
> Yes, here the rotation angles were sampled uniformly, and then filtered according to certain criteria (see Methods). The effect of rotation angle is investigated in detail (for networks only) in Abbas & Deny (2022), fig 5. They report: “We find that all networks are most fragile when the object is rotated by 90° in the out-of-plane condition (ObjectPose). We also find that the best networks are more robust across the full range of rotation angles in the in-plane and image-rotation conditions than in the out-of-plane condition.”
>
> **Given that the stimuli are artificially generated, introducing some textural confounders (similar to the aforementioned Geirhos et al. 2018 study) might also give a more nuanced and well-grounded picture.**
>
> This is indeed a good point and it would be interesting to study this question in a further study.

---

> ### Author Response · Authors · 2024-12-09
>
> **Further minor concerns: I wonder what kind of experiments the authors performed to make sure results are robust against certain reasonable variations of the setup. For example, why just pick one 3D model per object type? What about the text prompt provided to VLMs? For example, what happens if you conduct the experiment twice with the same VLM, but by consistently switching the choices of A & B, or by varying the text prompt?**
>
> We picked one object per class so that the humans could not use memory from previous trials to infer object identity, and because of the difficulty of finding multiple high-quality 3D models for one class.
>
> Swapping answers A and B did not change the accuracy of the VLMs. We ran all VLM analyses also with switched choices. That did not have any effect.
>
> Changing the prompt did affect the results, but we carefully selected the prompt that worked well for all VLMs.
>
> For example, we also tried the following prompt with the rotated-incorrect images:
>
> *The image represents either [label1] or [label2]. Which is it? Answer in one or two words.*
>
> For GPT-4o, this prompt led to 78\% correct answers, whereas our original prompt had 76\% correct answers. However, for Gemini, this prompt led to 91\% correct answers, against 95\% for our original prompt. This illustrates the variability that comes with changing the prompt.
>
> => We have now added this information to App A.
>
> **Deriving an answer to the two-forced choice question by simply looking at softmax activations and picking the bigger one largely makes sense, but should there perhaps be a cutoff in terms of how high the probability needs to be before accepting a correct answer? E.g. let's say that if for a certain stimulus, the DNN would assign to the correct label a probability of 0.01, and to the incorrect label a probability of 0.009, and to a third label entirely a probability of 0.9. Would this still count as a correct answer given that 0.01 is larger than 0.009, and the 0.9 probability answer is ignored?**
>
> This is a good point, but note that introducing such a cutoff would decrease the performance of the models. Thus, if anything, our protocol overestimates the ability of networks to provide the correct answer, and our conclusion that the pure vision models perform worse than humans (and some VLMs) would not change if such a cutoff was introduced.
>
> It is also good to keep in mind  that there is no such threshold in the experiment for humans (or VLMs). The human participants have to pick from the two alternatives, even if their first (however strong) percept would’ve corresponded to some altogether different object.
>
> We are investigating in details the question of calibration (network confidence) in a further study.

---

> > ### Author Response · Authors · 2024-12-19
> >
> > Dear reviewer, was there anything else we could do to help you reach a final recommendation? BR, Authors.

---

> > > ### Comment · Reviewer_ZRm5 · 2025-01-06
> > > **Positive (Conditional) Recommendation**
> > >
> > > After reading all reviews and author responses, I am happy to make a positive recommendation. I do have one main concern still (that should be addressed in the text) + some further comments below.
> > >
> > > Main concern:
> > > - effect of rotation axis on humans:
> > >   - I appreciate the summary of experiments in Abbas and Deny, and the authors make a fair point that it's difficult to do repeated trials with the same object in different rotations. However, is that really an obstacle to a more in-depth analysis for rotations? I think that it would have been possible to conduct such an analysis for some object classes for which you have multiple instances that look sufficiently different, or by allowing for variations in what you show to different human subjects. On the one hand, this study goes beyond the work of Abbas and Deny by conducting a nice comparison between DNNs and human subjects, but at the cost of a fair amount of detail in the analysis. More nuanced results would perhaps address the usual concerns that these results aren't actionable.
> > >   - I do however understand that this is not sth that can be addressed without a full repeat of the experiments + costly redesign, but I would really appreciate it if this point were to be discussed in the final manuscript in a Limitations section with some thoughts on how follow-up experiments could take this into account.
> > >
> > > Further comments:
> > > - effect of context clues:
> > >   - I fully agree with the authors that adding backgrounds would result in a less clean experimental protocol, and would introduce unwanted recognition cues. On the other hand, I maintain that experimenting with the object texture to tease out the effect of texture vs. shape would be useful, and I am glad that the authors are considering this for further study.
> > > - effect of language on recognition ability:
> > >   - The potential effect of language was raised in both other reviews, given that the artificial models with language-related components perform better than their vision-only counterparts. However, I suspect that there is a much more pedestrian explanation here, namely that VLMs can be trained on images that weren't explicitly annotated with a class label: This simply expands the scope of potential training images, since you can harvest a lot more data from the internet, and are also not restricted to images that correspond to some closed set of objects classes. I think for a solid conclusion on the effect of language, you would have to control for the type of label -- e.g. using the same set of images for two experiments, once with internet captions and once with class labels.
> > > - effect of calibration:
> > >   - The arguments against selecting a soft-max cut-off point are convincing, especially the lack of a threshold for humans.
> > > - effect of prompt:
> > >   - Thanks for adding the clarifications to Appendix A.

---

> > > > ### Author Response · Authors · 2025-01-06
> > > >
> > > > Thank you for your positive recommendation! We can certainly add the limitation about effect of rotation axis in the discussion and discuss follow-up experiments that would take this into account. We will update the paper in the coming days.

---

> > > > > ### Author Response · Authors · 2025-01-09
> > > > >
> > > > > We have now updated the "Discussion" section with a "Limitations" subsection.

---

> > > > > > ### Comment · Reviewer_ZRm5 · 2025-01-09
> > > > > > **Limitations Subsection**
> > > > > >
> > > > > > Thanks for the quick update. Adding an explicit discussion of limitations is a good first step.
> > > > > >
> > > > > > You mention three limitations in the new paragraph:
> > > > > > (1) context cues
> > > > > > (2) shape vs. texture.
> > > > > > (3) sampling rotation angles
> > > > > >
> > > > > > I have the following suggestions:
> > > > > >
> > > > > > (1) I would remove the point about context cues, because it's not a limitation imo. Adding natural object-specific backgrounds would only make it more difficult to measure robustness against pose changes. Perhaps I am missing something, but I don't see how adding this would strengthen your results. On the contrary even: Had you included natural backgrounds, I would have considered this to be a critical flaw in the setup.
> > > > > >
> > > > > > (2) I would move the point about shape and texture to what is now subsection 4.4 (after you cite Geirhos et al.), since that is where you speculate about the role of shape vs. texture. That would be a natural point to mention adopting parts of their setup for potential follow-up work, rather than mentioning it out of context in an earlier subsection.
> > > > > >
> > > > > > (3) That only leaves the third limitation re rotation angles, so I would thus dedicate this subsection to discussing possible future improvements when it comes to considering rotations at a more granular level. A few things came up during the discussion above that could be added (e.g. in your [comment](https://openreview.net/forum?id=yzbAFf8vd5&noteId=CiUGuvfdix) the discussion of insights from Abbas & Deny as well as Marr & Nishihara).

---

> > > > > > > ### Author Response · Authors · 2025-01-16
> > > > > > >
> > > > > > > We now provide an extensive discussion section dedicated to the effect of rotation angles and what could be done to explore them systematically. Copied below for your convenience.
> > > > > > >
> > > > > > > "\subsection{Main limitation of the current study}
> > > > > > >
> > > > > > > A critical extension of this study would be to systematically explore how viewpoint affects the performance of humans and networks. In \cite{Abbas_Deny_2023}, the effects of different rotation axes (image-plane rotation vs. 3D rotation) and rotation range are investigated in detail for networks. It is found that networks that have been trained with image rotation as a data augmentation scheme perform well on image-plane rotations, but are still impaired on 3D out-of-plane rotations. It is also found that networks perform at their worst on objects rotated 90° from their canonical pose, regardless of rotation axis. It would be interesting to study whether time-limited humans have the same or different failure modes, as this would shed light on their respective mechanisms.
> > > > > > >
> > > > > > > While our current study doesn't allow a systematic exploration of these factors, we can still comment on some of the typical failure modes of time-limited humans. It appears that many challenging images are such that the major axis of the object is occluded due to out-of-plane rotation, causing one end of the major axis to point towards the observer \citep{marr1978representation}. Examples include the corkscrew and the jellyfish in the rotated correct image (Figure \ref{human_errors}) as well as the violin and the crib in the rotated incorrect image (Figure \ref{human_Errors_2}). In other cases, it seems that occlusion of a signature detail of an object, such as the watch face or a lawnmower motor, also hinders human recognition performance. This also happens most frequently due to out-of-plane rotations. It is important to notice, however, that there are images that are challenging despite only involving in-plane rotation, such as the jellyfish and the parachute in the rotated incorrect image. Thus, a challenging rotation condition is often quite object specific.
> > > > > > >
> > > > > > > For human participants, it is difficult to obtain quantitative results of recognition as function of rotation for real objects, as each individual object cannot be presented to each observer in different rotation conditions (as there is heavy transfer of recognition across rotations). Different objects, on the other hand, will be differently affected by the same rotations. Artificial objects may be better stimuli for the quantitative characterization of the effect of rotation on human and network recognition ability, as have recently proposed \citep{bonnen2024evaluating}."
> > > > > > >
> > > > > > > Thank you again for your careful review.

---

### Review · Reviewer_PEXF · 2024-11-25

**Summary Of Contributions:**

This study investigates the performance of humans and neural networks in recognizing objects in challenging poses. The authors created a dataset of 147 images from 51 ImageNet categories, featuring objects in upright poses, "Rotated-Correctly" poses (rotated objects correctly classified by a pre-trained EfficientNet), and "Rotated-Incorrectly" poses (rotated object misclassified by the EfficientNet). Each image was paired with a two-choice question: the correct class and EfficientNet's top incorrect prediction.

Key experiments included tests on vision-only networks (SWAG, ViT, SWIN, BEIT, ConvNext), vision-language networks (Gemini 1.5, GPT-4o, and Claude 3), and humans under three conditions: (1) time-unlimited exposure, (2) 40ms time-limited exposure, and (3) 150ms time-limited exposure.

Their main observations are: Humans outperformed networks in recognizing objects under unusual poses in the time-unlimited condition. Vision-only networks and most vision-language models performed poorly on rotated objects, except for the Gemini models. When humans were restricted to 40ms exposure, their accuracy dropped to the level of networks. However, error patterns differed between humans and networks: from visual inspection it seems that humans struggled when the object structure was unclear while networks struggled when the object details (texture) were unclear.

**Audience:**

Yes

**Broader Impact Concerns:**

No concerns.

**Claims And Evidence:**

Yes

**Requested Changes:**

- While the study evaluates several vision-only networks, selected based on their performance in a previous related study, it would have been more complete to also include more recent and widely used models. For example, the state-of-the-art self-supervised model DINOv2 [A] and vision-contrastive models like CLIP [B] and its newer variations (EVA-CLIP [C] and SigLIP [D])are among the most popular image encoders today. Adding these models would give a broader view and better reflect the capabilities of current image encoders.  For instance, these models are often tested in recent research on robustness for tasks like semantic segmentation [F, G].

- The analysis would have been richer if the work was also examining how the following factors affect the performance of the networks: a) the size of the pre-training dataset, b) the model size, c) the pre-training objective, and d) the resolution or size of the input images. Although the evaluated networks vary in these aspects, the study does not clearly address whether or how these factors affect performance. Such deeper investigation into these dimensions might have provided more insights on how to address the "brittleness" of vision-only networks on recognizing objects under challenging poses. For example, it would be informative to understand whether increasing both model size and pre-training dataset size increase the robustness of the models.

- Gemini's behavior diverges so significantly from the other evaluated models. This raises the question of whether there is a possibility that the dataset used for evaluation in this study might have been included in the training data for such closed-source models. These models often rely on extensive internet-scraped datasets, which makes it hard to know exactly where the data came from. Do the authors consider this a possibility? What I mean is that, while it's nearly impossible to confirm such overlap, it would be helpful to clarify whether the images in this study were available online and could have been included in the training data of these models, which gather images from the internet.

- Building on the previous point, it would have been valuable to include open-source and better-documented vision-language models, such as LLaVA 1.5 [E], in the evaluation. LLaVA 1.5 typically uses a CLIP-like model as its image encoder, and testing it in this context could provide insights into whether adding a language decoder (as LLaVA does) enhances performance on objects in challenging poses.


[A] DINOv2: Learning robust visual features without supervision, In TMLR 2024
[B] Learning transferable visual models from natural language supervision, In ICML 2021
[C] EVA-CLIP: Improved training techniques for clip at scale.
[D] Sigmoid loss for language image pre-training, In ICCV 2023.
[E] Improved Baselines with Visual Instruction Tuning, In CVPR 2024
[F] How to Benchmark Vision Foundation Models for Semantic Segmentation?, In CVPR 2024 workshop.
[G] Stronger, Fewer, & Superior: Harnessing Vision Foundation Models for Domain Generalized Semantic Segmentation, In CVPR 2024

**Strengths And Weaknesses:**

This study provides insights into human visual perception, the limitations of vision and vision-language networks, and the differences between the two (see insights/observations in the summary section). It contributes to understanding vision systems and suggests potential ways to enhance network performance, such as introducing recurrent processes.

The study also reveals a notable performance gap between vision-only networks and vision-language models like Gemini and GPT-4, with Gemini showing particularly strong accuracy on objects in unusual poses. However, the reasons for this difference remain unclear (partly due to limited documentation on these systems).

---

> ### Author Response · Authors · 2024-12-09
>
> Dear Reviewer,
>
> Thank you for your review and valuable comments, to which we answer below:
>
> **Trying DINOv2 [A] and vision-contrastive models like CLIP [B] and its newer variations (EVA-CLIP [C] and SigLIP [D])**
>
> Thank you for the suggestions.
>
> => We have now evaluated SigLip and DINOv2 on our dataset.
>
> SigLip:
> 	upright condition:  98.03922%;  rotated 83.99%
>
> DinoV2:
> 	upright condition: 100%; 	rotated 46.34%
>
> We have now added SigLIP to the paper in the category of VLMs. We decided not to add DinoV2 because it does not qualify as a state-of-the-art model on this task.
>
> CLIP (zero-CLIP_ViT_B_ens (400M, 86M)) was tested in Abbas & Deny (2022) on a very similar dataset and obtains:
> upright: 95% ; 	rotated: 57%.
> We suspect that it under-performs other models because it was not fine-tuned on ImageNet (63.2% accuracy on ImageNet). We do not add it to our comparison here because it does not qualify as a very good model on this task.
>
> We encoutering technical difficulties testing EVA-CLIP for now (model-hub not reachable).
>
> *Abbas & Deny (2022) : Progress and limitations of deep networks to recognize objects in unusual poses https://arxiv.org/abs/2207.08034*
>
> **Examining how the following factors affect the performance of the networks: a) the size of the pre-training dataset, b) the model size, c) the pre-training objective, and d) the resolution or size of the input images.**
>
> The factors (a) to (c) (size of pre-training dataset, model size, pre-training objective) have been investigated very thoroughly on a collection of 37 models including the ones we test here, on a very similar dataset, in Abbas & Deny (2022). (e.g see fig 2.A and Table 1). They find: “We show that classifying these images is still a challenge for all networks tested, with an average accuracy drop of 29.5% compared to when the objects are presented upright. This brittleness is largely unaffected by various network design choices, such as training losses (e.g., supervised vs. self-supervised), architectures (e.g., convolutional networks vs. transformers), dataset modalities (e.g., images vs. image-text pairs), and data-augmentation schemes. However, networks trained on very large datasets substantially outperform others, with the best network tested—Noisy Student EfficentNet-L2 trained on JFT-300M—showing a relatively small accuracy drop of only 14.5% on unusual poses.”
>
> For this study we selected the 5 best networks on unusual poses based on Abbas & Deny (2022). The goal was to compare the average accuracy of the very best networks available on this task, to humans. The small number of networks used in this study however is not ideal to run the type of thorough analysis that was run in Abbas & Deny (2022).
>
> =>We added details about the architecture and pre-training datasets for each model, in Appendix C.
>
> **Gemini's behavior diverges so significantly from the other evaluated models. This raises the question of whether there is a possibility that the dataset used for evaluation in this study might have been included in the training data for such closed-source models.**
>
> Indeed, our 3D models are collected online from sketchfab.com and it is completely possible that Gemini was trained on the same data (data contamination). We now say: “The impressive performance of vision-language models suggests that they might be able to exploit similar mechanisms for image recognition as humans, which could for instance involve retro-fitting the labels to the image. However, an alternative explanation is that they could have been fed a different data diet involving more rotated objects than their pure vision counterparts. Indeed, Abbas & Deny (2022) found that increasing the training dataset size generally resulted in better performance in rotated object recognition. *The exceptional performance of Gemini could even be due data contamination issues, since the objects we selected were available online.* Because of the limited documentation available for these models, it is difficult to come to a firm conclusion regarding the mechanisms underlying their abilities.”
>
> =>We added the point about data contamination in the discussion.
>
> **it would have been valuable to include open-source and better-documented vision-language models, such as LLaVA 1.5 [E], in the evaluation.**
>
> This is a great point, and we should include LLaVA in our future studies, but for the moment we have encountered technical difficulties to making it work (the specific required library bitsandbytes was not possible to install on our current available hardware).

---

> > ### Comment · Reviewer_PEXF · 2024-12-12
> > **Response to Authors**
> >
> > Thank you to the authors for their detailed responses. I appreciate the effort in addressing my concerns and confirm that I reviewed their responses before making my final recommendation.

---

### Decision · Action_Editor_nZPj · 2025-01-06

**Recommendation:** Accept with minor revision

**Comment:**

I think this paper clearly meets the standards of claims and audience. I'd ask the authors to make sure the limitations section of the final version addresses the points raised by reviewer ZRm5 as needed.

**Audience:**

The reviewers agree (and I concur) that this paper provides an interesting study. Given the importance of robust object perception for the increasingly many grounded applications of AI, this paper will clearly be of interest for practical applications, as well as the science of understanding model generalization.

**Claims And Evidence:**

This paper studies the robustness of object recognition from uncommon perspectives in language models and humans. The results find that most models are less robust than humans, though human performance on challenging stimuli requires additional integration time, suggesting additional mechanisms are utilized in these cases. The reviewers agree (and I concur) that the experiments generally support these claims.

---

> ### Author Response · Authors · 2025-01-16
>
> Thank you for this positive decision, and we would like to thank once more the reviewers for their careful and insightful reviews of our work, and the AC for this smooth reviewing process. We have now addressed the last concerns of reviewer ZRm5 regarding the limitations section to the best of our ability.

---

> > ### Author Response · Authors · 2025-01-17
> >
> > Camera ready version posted. Thanks all.